# A modular platform for bioluminescent RNA tracking

Lila P. Halbers [1,5], Kyle H. Cole[2,5], Kevin K. Ng[1,5], Erin B. Fuller [3], Christelle E. T. Chan [1], Chelsea Callicoatte[4], Mariajose Metcalfe[4], Claire C. Chen[1], Ahfnan A. Barhoosh[1], Edison Reid-McLaughlin[3], Alexandra D. Kent[3], Zachary R. Torrey[3], Oswald Steward[4] ✉, Andrej Lupták [1,2,3] ✉ & Jennifer A. Prescher [1,2,3] ✉

A complete understanding of RNA biology requires methods for tracking transcripts in vivo. Common strategies rely on fluorogenic probes that are limited in sensitivity, dynamic range, and depth of interrogation, owing to their need for excitation light and tissue autofluorescence. To overcome these challenges, we report a bioluminescent platform for serial imaging of RNAs. The RNA tags are engineered to recruit light-emitting luciferase fragments (termed RNA lanterns) upon transcription. Robust photon production is observed for RNA targets both in cells and in live animals. Importantly, only a single copy of the tag is necessary for sensitive detection, in sharp contrast to fluorescent platforms requiring multiple repeats. Overall, this work provides a foundational platform for visualizing RNA dynamics from the micro to the macro scale.

RNA dynamics play pivotal roles in a multitude of cellular processes[1]. While we have a deep, molecular-level understanding of many facets of RNA biology in vitro, the picture in physiologically authentic environments—live animals—remains incomplete. This is due, in part, to a lack of methods for noninvasive tracking of RNAs in vivo. Conventional approaches rely on RNA tags coupled with fluorescent probes[2–5]. Such platforms require excitation light, which can be difficult to deliver in whole organisms without invasive procedures, excision of tissues, or delivery of fluorogenic dyes[6–8]. Furthermore, external light can induce autofluorescence, precluding sensitive detection of low-abundance targets. Short imaging times are also necessary to avoid light-induced damage. Consequently, tracing the lifecycle of key RNAs in real time, in live mammals, has been elusive.

We reasoned that a potentially more suitable platform for RNA imaging in live animals could be achieved with bioluminescence. This modality relies on photon production from luciferase enzymes and luciferin small molecules. Since no excitation light is required, bioluminescence can provide superior signal-to-noise ratios in vivo for visualization of low-copy transcripts. Additionally, serial imaging is possible without concern for phototoxicity or tissue damage. The development of luciferases with higher photon outputs and improved thermal stability (e.g., NanoLuc and related variants) has enabled facile visualization of cells, biomolecules, and other features both on the micro and macro level[9–12]. Recently, a split variant of NanoLuc was applied to RNA targets, setting the stage for precise detection of cellular transcripts[13].

Here we report a general method that leverages advances in bioluminescence technology for multi-scale RNA detection. The approach features split fragments of NanoLuc (NanoBiT) fused to MS2 and PP7 bacteriophage coat proteins (MCP and PCP), fusions that we have termed RNA lanterns. MCP and PCP bind distinct RNA aptamers (MS2 and PP7, respectively) that can be appended to transcripts of interest[14,15]. Upon transcription, MCP and PCP bind the MS2-PP7-containing RNA bait, bringing the luciferase fragments into proximity to assemble a functional, light-emitting enzyme. We extensively optimized both the lanterns and RNA bait to maximize signal turn-on and

[1]Department of Pharmaceutical Sciences, University of California, Irvine, Irvine, CA, USA. [2]Department of Molecular Biology and Biochemistry, University of California, Irvine, Irvine, CA, USA. [3]Department of Chemistry, University of California, Irvine, Irvine, CA, USA. [4]Department of Anatomy and Neurobiology, University of California, Irvine, Irvine, CA, USA. [5]These authors contributed equally: Lila P. Halbers, Kyle H. Cole, Kevin K. Ng. ✉e-mail: osteward@uci.edu; aluptak@uci.edu; jpresche@uci.edu

minimize the size of the protein-RNA complex. Notably, a single rigidified RNA was sufficient for sensitive imaging both in cells and in vivo, rendering much larger RNA tags unnecessary. Additionally, the RNA bait is modular and can be used in conjunction with other split luciferases and for multi-scale imaging. The tools reported here are thus immediately useful to studies of RNA dynamics.

## Results

### General strategy for visualizing RNAs with bioluminescent light

The overall strategy was dependent on three key steps: RNA bait formation, MCP/PCP binding, and NanoBiT complementation. We took inspiration from MCP- and PCP-based reporters comprising split fluorescent proteins to visualize transcripts[16]. We envisioned that the RNA-binding proteins could be fused to the NanoBiT system[17], which has been used extensively to examine protein-protein interactions and other biomolecular networks[18-20]. We appended each part of NanoBiT (the short peptide—SmBiT—and the larger protein fragment—LgBiT, Fig. 1A) to MCP and PCP, respectively. The resulting fusions (termed RNA lanterns) were hypothesized to bind their cognate aptamers on a single transcript, inducing NanoBiT complementation and generating photons in the presence of luciferin. The requisite components had not been used together prior to this work, necessitating optimization of each step.

As a starting point, we designed a bicistronic construct with an internal ribosome entry site (IRES) to mediate co-expression of the RNA lantern components (Fig. 1A). MCP was fused to SmBiT and PCP was fused to the larger protein fragment (LgBiT), building on a previously published split fluorescent protein platform, in which MCP and PCP were fused to the N- and C-terminal fragments of the Venus fluorescent protein, respectively[16]. MCP was further tagged with a nuclear localization signal (NLS) to reduce background complementation. This same strategy was used previously for the split Venus system[16] to separate the lantern fragments in the absence of the RNA bait. Upon expression, transcripts fused to the RNA bait could transport an MCP-fusion from the nucleus into the cytoplasm (or capture de novo translated MCP in the cytoplasm). Eventual binding of the PCP fusion would co-localize both halves of the RNA lantern on the target transcript, enabling NanoBiT complementation and thus light production. Additionally, the NLS reduces the likelihood of non-specific SmBiT/LgBiT complementation, a problem encountered in previous studies that resulted in diminished sensitivity[13].

We modeled the RNA lanterns using ChimeraX[21] to assess the design of the fusions (Fig. 1B). No obvious steric clashes or unfavorable orientations with a 5′-MS2-PP7-3′ RNA bait were observed, suggesting that lantern assembly was possible. NanoBiT complementation appeared feasible even with juxtaposed MS2 and PP7 aptamers. Small, compact

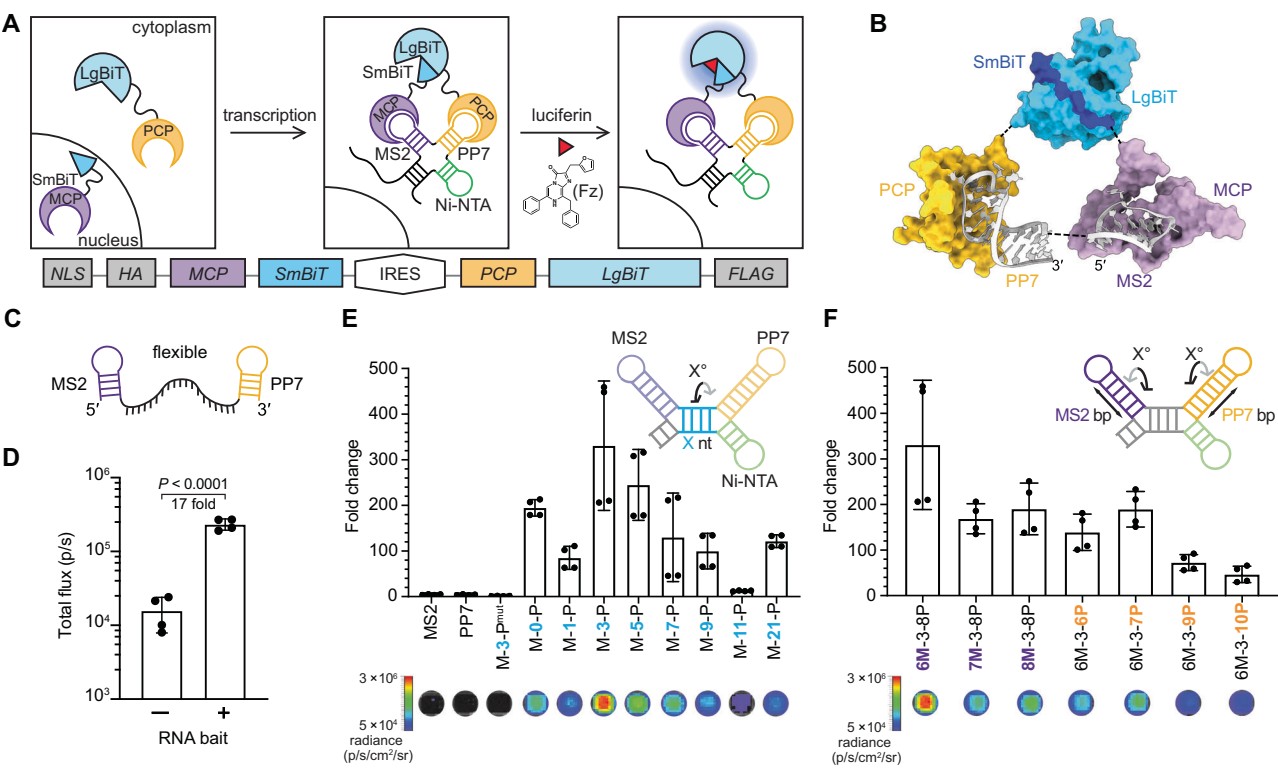

**Fig. 1 | An optimized platform for tracking RNA dynamics. A** Strategy to visualize transcripts using RNA lanterns. The lanterns comprise fusions of MS2 coat protein (MCP) and PP7 coat protein (PCP) with NanoBiT fragments (SmBiT and LgBiT, respectively). Transcription of bait RNA (comprising MS2 and PP7 aptamers) drives NanoBiT heterodimerization. In the presence of furimazine (Fz), light is produced ($\lambda_{max}$ = 460 nm, blue glow). The bicistronic construct encoding the RNA lantern is shown below the scheme. MCP and PCP were fused with HA and FLAG tags, respectively, for expression analyses. **B** Modeling of the RNA lantern complex. Crystal structures of MS2 (purple, 1ZDI)[43], PP7 (orange, 2QUX)[44], and NanoLuc (5IBO), highlighting LgBiT (dark blue) and SmBiT (cyan), were modeled in ChimeraX[21]. The MCP-SmBiT/PCP-LgBiT complex was modeled by aligning the corresponding N- and C- termini of each protein and aligning the 5′ and 3′ ends of the aptamers. **C** Predicted secondary structure of flexible RNA bait, as calculated by RNAfold[22]. **D** RNA lantern with a flexible RNA bait via an in vitro transcription and translation (IVTT) assay. The total flux observed ± RNA bait. $P = 4.16 \times 10^{-5}$ (95% confidence interval) calculated by a two-sided $t$-test. **E** Engineered rigid RNA baits provided robust photon output in IVTT assay. RNA baits containing varying spacers between the MS2 and PP7 aptamers were constructed, with spacer length (X nt, blue, denoted below each bar graph). Fold change in signal over RNA lantern alone is plotted. MS2, PP7, and M-3-P$^{mut}$ denote baits comprising isolated aptamers or a mutated PP7 aptamer (Supplementary Fig. 3A, B), respectively, none of which were expected to result in RNA lantern assembly. Representative luminescence images are shown below the graph. **F** Rigid RNA baits comprising various MS2 and PP7 stem lengths. Luminescence readouts were acquired following IVTT. Fold change in signal over RNA lantern alone is plotted. Representative luminescence images are shown below the graph. Data are presented as mean values ± SD for $n$ = 4 replicates (**D**, **E**, and **F**). Source data for (**D**, **E**, and **F**) are provided as Source Data files.

baits are attractive tags to avoid disrupting the structures or functions of target RNAs. Encouraged by the modeling results, we moved forward with the lantern design and anticipated that additional engineering of the lantern components (linkers and orientation) would be necessary to maximize luciferase complementation and signal output.

## Biochemical optimization of the RNA tag and lanterns

The RNA detection platform relies on efficient NanoBiT formation, which can be tuned based on the SmBiT peptide sequence[17]. We examined two SmBiT peptides (SmBiT$^{high}$, $K_D = 180$ nM; SmBiT$^{low}$, $K_D = 190$ µM) to determine which would provide the best dynamic range: minimal signal in the absence of RNA bait and robust signal in its presence (Supplementary Fig. 1A). The designer probes were first expressed using an in vitro transcription/translation system (IVTT) featuring transcription by T7 RNA polymerase and translation in rabbit reticulocyte lysate (RRL). Background signal was determined in the absence of RNA bait. Full complementation of translated NanoBiT was achieved by adding exogenous SmBiT$^{ultra}$ ($K_D = 0.7$ nM) or recombinant LgBiT at saturating concentrations, establishing the maximum potential signal (Supplementary Fig. 1B). All protein designs exhibited robust signal enhancement when SmBiT$^{ultra}$ or recombinant LgBiT was added, but high background luminescence was observed in samples with SmBiT$^{high}$. The largest signal enhancements were achieved with SmBiT$^{low}$, due to the reduced background complementation observed with this peptide (Supplementary Fig. 1C). Additional repeats of SmBiT did not yield greater light output, possibly due to insufficient spacing between sequential peptides to support LgBiT binding. We therefore moved forward with MCP-SmBiT$^{low}$/PCP-LgBiT, the lantern combination that provided the highest dynamic range.

We first asked whether the MCP-SmBiT$^{low}$/PCP-LgBiT lantern could detect the previously reported MS2-PP7 bait, comprising an unstructured 19-nucleotide strand joining the two aptamers (Fig. 1C)[14]. Upon RNA bait transcription, an approximate 17-fold signal increase was observed (Fig. 1D). We attributed the relatively low signal enhancement to the RNA bait structure. Secondary structure modeling with Vienna RNAfold[22] and Forna[23], revealed that the 19-nucleotide linker was likely flexible with the potential to sample non-productive conformations. While an unstructured RNA bait could aid lantern binding, we surmised that extensive flexibility could potentially hinder NanoBiT complementation and thus diminish signal-to-noise ratios.

We hypothesized that increasing the rigidity of the RNA bait would favor lantern assembly and photon production. We thus redesigned the RNA bait to fix the spacing and orientation of the MS2 and PP7 sequences. We locked the aptamers into desired conformations, as part of a four-way junction with an established Ni-NTA-binding aptamer (Fig. 1E and Supplementary Fig. 2A)[24]. The RNA baits were further engineered to contain varying numbers of nucleotides between the MS2 and PP7 aptamer domains (M-X-P; where X is the number of nucleotides in the spacer). This panel enabled us to not only sample the spacing between the aptamer domains (~3 nm, according to secondary structure predictions) but also the helical phase of the aptamers respective to one another (up to ~360°). When the RNAs were present with the lantern fragments, we observed that a relatively short linker (M-3-P) could yield significantly higher luminescence, with up to a 330-fold increase over a no-RNA control (Fig. 1E). In the case of M-11-P, signal was abolished. We attributed this result to the phase angle of ~180° (one-half of a helical turn) from M-3-P, preventing luciferase complementation. Importantly, signal was restored with the MS2 and PP7 aptamers brought back into phase (~360°) and a full helical turn apart (M-21-P). Signal still decreased because the overall length increased and likely prevented the effective assembly of the luciferase parts. Photon production was also highly dependent on the concentrations of the RNA bait and the RNA lantern (Supplementary Fig. 2B) and RNA bait integrity. In experiments using singular MS2 and PP7 aptamers, or M-3-P$^{mut}$ that does not bind the PCP protein

(Supplementary Fig. 3)[25], no appreciable signal over background was observed (Fig. 1E and Supplementary Fig. 3C).

Due to the improved luminescence achieved through modulating the distance and phase angle, we also examined the length and relative orientation of the individual MS2 and PP7 aptamers. We took the optimal M-3-P RNA bait and added or removed base-pairs (bp) to either the MS2 or PP7 stem (Fig. 1F and Supplementary Fig. 4A). No significant improvements in photon output were observed among the suite of RNA baits, suggesting that the length and orientation of the two aptamers were already optimal. We further confirmed that the lack of improvement was not a result of template DNA ratios (Supplementary Fig. 4B). Although the orientation of the aptamers had minimal effect on complementation, we found that the positioning of the aptamers was critical. When the placement of MS2 and PP7 were inverted, luciferase complementation was not observed (Supplementary Fig. 5).

We further examined the effect of protein linker length on RNA lantern assembly. Constructs were designed with additional glycine-serine (G$_4$S) units inserted between the RNA-binding proteins and NanoBiT segments (Supplementary Fig. 6A). These constructs were then tested with the M-3-P, M-7-P, and M-11-P RNA baits (Supplementary Fig. 6B). The largest light outputs were achieved with the M-3-P RNA bait, for all the lanterns tested. Interestingly, only minimal signal enhancements (~1.5 fold) were observed with the additional G$_4$S units compared to the original lanterns (Supplementary Fig. 6C).

The RNA sensing capabilities of the designer bait and lanterns were biochemically validated in vitro (Fig. 2A). Cells were engineered to stably express the lantern components, and lysates were then titrated with purified bait. As shown in Fig. 2B-C, an RNA-dependent "hook effect" was observed (Fig. 2B, C)—increased amounts of RNA resulted in lantern fragments bound to separate transcripts and thus reduced light emission. Lantern complex formation was also limited at low RNA concentrations, as expected. More lantern complementation was observed with increasing RNA bait, but the amount falls off at high RNA concentrations. This latter outcome is explained by MCP and PCP proteins binding distinct RNA bait transcripts, so that the luciferase bits are not brought together to assemble active lantern complexes.

The unique design of the RNA bait enabled direct interrogation of lantern binding and assembly. The Ni-NTA aptamer was used to retrieve the RNA bait-RNA lantern complex on resin: as shown in Fig. 2D, the assembled NanoBiT enzyme was exclusively associated with the RNA bait (Fig. 2D). Pulldowns of active complexes using the HA and FLAG tags built into the RNA lantern components further confirmed the RNA-dependent assembly of the active luciferase (Fig. 2E, F). We anticipate that the retrieval of targeted transcripts and associated biomolecules post-imaging will facilitate the often-critical follow-up analyses of RNA interactions.

Real-time imaging of RNAs in cells demands fast signal turn-on in the presence of target transcripts. We thus evaluated the kinetics of lantern assembly in vitro. M-3-P bait (1 nM) was first added to lysate from lantern-expressing cells, and luminescence measurements were collected over time (Fig. 3A). Signal was immediately detected upon RNA addition, with peak luminescence reached within 20 min. The rapid reconstitution kinetics are on par with the most optimized split fluorescent reporters[26]. No luminescence above background was observed in the absence of RNA or when control M-3-P$^{mut}$ bait was added to the lysate. The lanterns can also report on gene expression in real-time. The split luciferase fragments were expressed in reticulocyte lysate prior to the addition of either purified M-3-P RNA bait, M-3-P DNA, or DNA encoding *GFP-M-3-P* (Fig. 3B). As expected, signal was immediately observed from samples containing the RNA bait. The onset of luminescence was delayed, though, for samples comprising the DNA bait, owing to the need for target transcription prior to lantern binding. The longest delay was observed with *GFP-M-3-P*, the construct with the largest sequence upstream of M-3-P. Collectively, these data suggest that the RNA lantern and rigidified tags can provide reliable readouts on RNA dynamics.

## RNA imaging platform is ultrasensitive and modular

Previous applications of MS2-PP7 for fluorescence imaging have required at least 12 copies of the aptamers to achieve adequate signal-to-noise ratios (MP$_{12X}$, Fig. 4A)[2,16,27,28]. In many cases, these constructs comprise an RNA bait that is at least 780 nucleotides long, a size that may impede the natural localization and behavior of the RNA under study[29–31]. The M-3-P bait (69 nucleotides) requires only a single copy of each aptamer to produce detectable signal (Fig. 4B). When compared

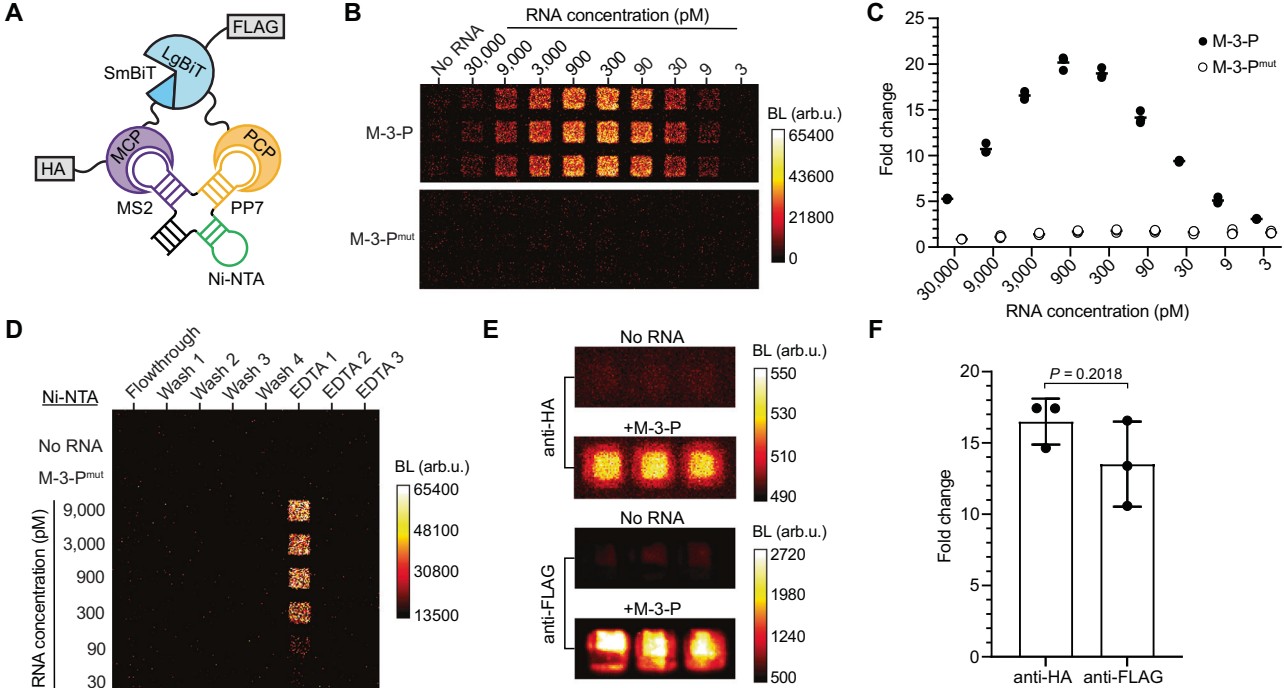

**Fig. 2 | Biochemical validation of RNA lanterns. A** The designer RNA bait comprises a 4-way-junction of MS2, PP7, and Ni-NTA aptamers. This unit assembles the RNA lantern components (MCP-SmBiT and PCP-LgBiT, fused to HA and FLAG epitopes, respectively). **B** Bioluminescent output from RNA bait (M-3-P) or inactive mutant (M-3-P$^{mut}$) combined with lysate from cells stably expressing the RNA lantern. **C** Fold change in bioluminescent signal over no-RNA controls. Each bar represents the mean of $n = 3$ replicates with dots showing data from individual replicates. **D** Affinity purification of the RNA lantern complex using the Ni-NTA aptamer. Various concentrations of RNA bait were used, and captured complexes were eluted using metal chelator (EDTA). **E** Bioluminescent output of lantern complexes captured using anti-HA or anti-FLAG conjugate-agarose beads. **F** Fold change in signal from lantern complexes retrieved in the presence of M-3-P bait versus no RNA from (**E**) $P$ (95% confidence interval) values were calculated using a two-sided $t$-test. BL bioluminescence, arb.u. arbitrary units. Data are presented as mean values ± SD for $n = 3$ replicates. Source data for (**B**–**F**) are provided as Source Data files.

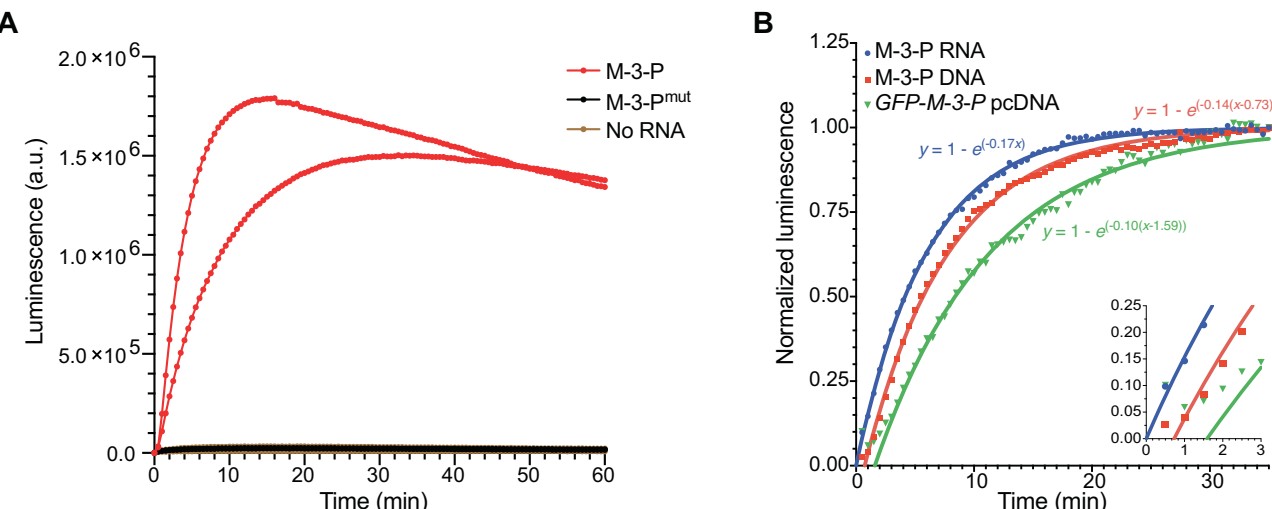

**Fig. 3 | Kinetics of RNA lantern assembly. A** Bioluminescent output from RNA bait (*M-3-P*, red), inactive mutant (*M-3-P$^{mut}$*, black), or no bait (gold). Samples were mixed with lysate from cells expressing RNA lanterns and analyzed over time. Two replicate experiments were performed for each condition and all runs are shown. **B** Real-time RNA lantern assembly with purified RNA bait, along with RNAs produced via in vitro transcription. Lanterns were expressed by in vitro transcription and translation in reticulocyte lysate prior to addition of either purified *M-3-P* RNA bait (1 nM), *M-3-P* DNA (1 nM; 69-nt transcript), or *GFP-M-3-P* DNA (1 nM; 880-nt transcript). Luminescence was measured every 30 s, and mean luminescence values (from $n = 3$ replicates) are plotted. Data were normalized at the 35-min timepoint for each 30 s interval. Solid curves are data fit to the mono-exponential function $y = 1 - e^{-k(t-t_0)}$, where $t_0$ represents a delay due to RNA production. Luminescence is detectable within 30 s of RNA introduction and half-maximum signal is observed in 4 min. Source data for (**A** and **B**) are provided as Source Data files.

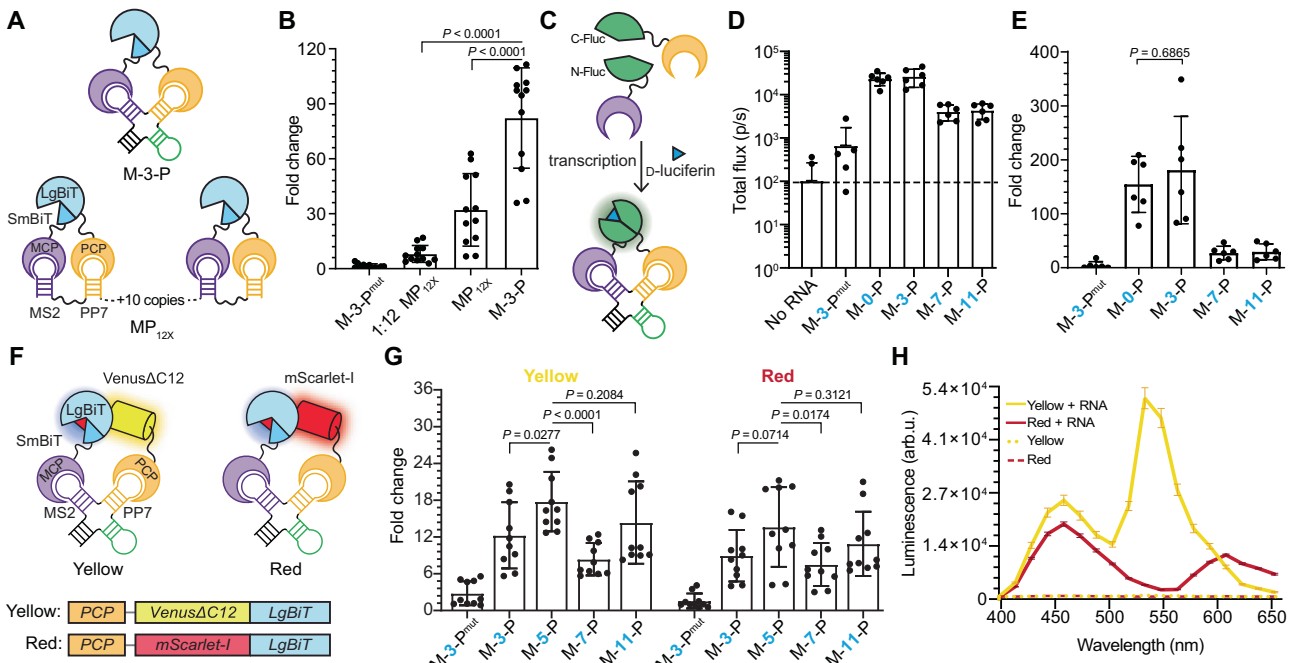

**Fig. 4 | Robustness and modularity of structured RNA baits. A** Comparison of M-3-P to a 12-copy unstructured RNA bait. Top: Schematic of the structured M-3-P RNA bait (single copy). Bottom: Schematic of a flexible RNA bait comprising 12 copies of the MS2 and PP7 aptamers (MP₁₂ₓ). **B** MP₁₂ₓ was evaluated against M-3-P at equimolar RNA concentrations (1 nM of template DNA) $P = 6.10 \times 10^{-13}$ or at 1/12 the concentration (83 pM of template DNA) $P = 4.77 \times 10^{-9}$. Fold change over no-RNA bait samples is plotted. Data are presented as mean values ± SD. **C** RNA bait assembles other lanterns. Schematic of MCP-PCP probes fused to split firefly luciferase (Fluc). MCP is fused to the N-terminal half of Fluc and PCP is fused to the C-terminal half. RNA transcription followed by D-luciferin treatment enables photon production and visualization ($\lambda_{max} = 560$ nm, green glow). **D** The Fluc lantern was assessed with a panel of RNA baits. Total light output for $n = 6$ replicates is shown. Only two replicates are shown for the no-RNA bait sample; the others were

below the limit of detection represented by the electronic noise of the EMCCD camera (dashed line). Data are presented as mean values ± SD. **E** Graph of fold change in signal over no-RNA bait samples from (**D**) using a two-sided *t*-test. Data are presented as mean values ± SD. **F** Schematic of BRET-based RNA lanterns. **G** Luminescence fold change measurements for yellow and red RNA lanterns across a panel of RNA baits. Fold change calculated over no-RNA controls. $P = 4.05 \times 10^{-5}$ for yellow M-5-P vs M-7-P. Data are presented as mean values ± SD. **H** Emission spectra for BRET-based RNA lanterns. Error bars represent the standard deviation (SD) for $n = 12$ replicates in (**B**), $n = 6$ replicates in (**D** and **E**), $n = 10$ replicates in (**G**), and $n = 3$ replicates in (**H**). arb.u., arbitrary units. The *P* values (95% confidence interval) in (**B**, **E**, and **G**) were calculated by a two-sided *t*-test. Source data for (**B**, **D**, **E**, **G**, and **H**) are provided as Source Data files.

head-to-head with bait aptamers at equimolar concentration (M-3-P versus one twelfth RNA concentration of MP₁₂ₓ), the structured bait produced 10-times more signal. Even when MP₁₂ₓ was introduced at the same RNA concentration and the aptamers were 12-times more concentrated, the structured M-3-P bait outperformed the flexible design (Fig. 4B). These results suggest that the compact M-3-P is a more effective RNA bait, providing higher signal compared to the multimerized and extensively used MS2-PP7 tag.

The designer M-3-P bait can also facilitate complementation of other split reporters. When MCP and PCP were fused to split fragments of firefly luciferase (Fluc; Fig. 4C), the resulting RNA lantern yielded a large increase in signal over background (Fig. 4D, E). The photon flux of the assembled Fluc is lower than NanoBiT, but the undetectable background in the absence of RNA may be desirable in specific applications. M-3-P also enabled the assembly of bioluminescence resonance energy transfer (BRET) variants of NanoLuc. These probes comprise luciferase fluorescent protein fusions that produce red-shifted light upon complementation. Such wavelengths can provide more sensitive readouts in vivo because they are less absorbed and scattered by tissue. Two BRET-based RNA lanterns were developed using yellow (VenusΔC12)[10] and red (mScarlet-I)[32] fluorescent proteins fused to the C-terminus of PP7 and the N-terminus of LgBiT (Fig. 4F). Assembly of the yellow and red RNA lanterns was assessed by measuring photon production in the presence of rigidified RNA baits. As shown in Fig. 3G, M-5-P provided the highest degree of signal turn-on (Fig. 4G). The larger bait provides more space between the bulkier lantern fragments in these cases, likely enabling more effective

complementation. The BRET-based probes also exhibited different colors of light output (Fig. 4H). BRET efficiency was higher for the yellow than the red lantern, likely due to the larger spectral overlap between NanoBiT and VenusΔC12 compared to mScarlet-I[10,33]. Collectively, these demonstrate that structured RNA baits can productively assemble diverse split proteins.

## Imaging RNAs from the micro-to-macro scale

The robustness of the bait design enabled facile imaging of RNA both in cellulo and in vivo. As an initial test, we developed a model system using an mRNA encoding green fluorescent protein (GFP) with the M-3-P RNA bait placed in the 3' untranslated region (UTR) (Fig. 5A). GFP fluorescence would thus report on bait expression. Cell lines stably expressing RNA lanterns were generated and transfected with constructs encoding M-3-P-tagged *GFP* or *GFP* only. GFP fluorescence was observed in both cases, but bioluminescence (from RNA lantern assembly) was only observed in cells transfected with M-3-P-tagged *GFP*. Similar signal turn-on was observed using an analogous model transcript. (Fig. 5B, C, Supplementary Fig. 6D, E, Supplementary Fig. 7, Supplementary Fig. 8, and Supplementary Fig. 9A) It should be noted, though, that many fluorescent cells were devoid of luminescence. Lantern expression can vary between cells, and bioluminescent signal is further dependent on target transcript levels and luciferin availability. RNA lifetimes vary from experiment to experiment, and bioluminescent signal will decrease if substrate is not replenished over time. All of these parameters must be considered in a given imaging study.

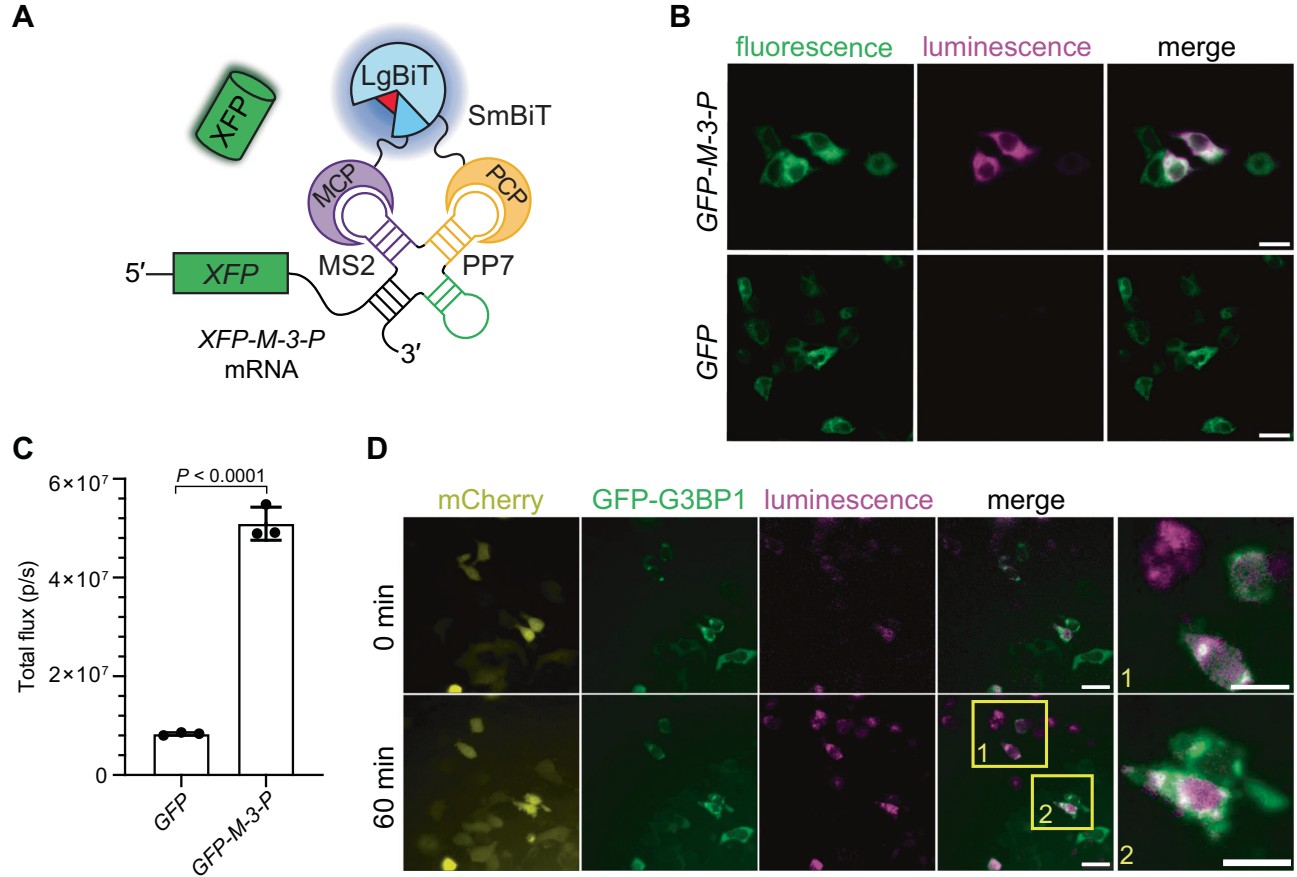

**Fig. 5 | Dynamic imaging in mammalian cells with RNA lanterns. A** Schematic of model mRNA encoding a fluorescent protein. The sequence was engineered with M-3-P in the 3' UTR. Transcription of *XFP-M-3-P* mRNA results in lantern assembly and light emission (blue glow). XFP production can be further analyzed via fluorescence (green glow). (**B**) HEK293T cells expressing the RNA lantern were transiently transfected with DNA encoding *GFP−M-3-P* (Pearson's $r = 0.84$) or *GFP* alone (Pearson's $r = 0.03$). Luminescence was observed exclusively in cells containing mRNAs with the M-3-P bait. Scale bar = 20 μm. **C** Bulk measurement of photon flux from lantern-expressing cells transfected with DNA encoding *GFP* or *GFP−M-3-P*.

$P = 2.55 \times 10^{-5}$ (95% confidence interval) calculated by a two-sided *t*-test. Data are presented as mean values ± SD for $n = 3$ replicates (**D**) Dynamic mRNA imaging. HEK293T cells expressing RNA lanterns were transfected with *mCherry-β-actin* (with M-3-P located in the 3' UTR) and GFP-G3BP1. GFP-G3BP1 is known to localize to stress granules[45,46]. Fluorescence readouts (mCherry) confirmed successful reporter transfection and expression. Cells were treated with sodium arsenite and imaged before (0 min) and after treatment (60 min), and $n = 6$ trials were conducted. Magnified views of two cell clusters are also shown. Scale bars = 20 μm. Source data for (**B**, **C**, and **D**) are provided as Source Data files.

The number of assembled lanterns in cells was further quantified via pulldown assays. RNA bait-RNA lantern complexes were retrieved via the Ni-NTA aptamer as before (Supplementary Fig. 9B). The amount of captured complex (assessed via luminescence measurements) correlated with the amount of target DNA and downstream mRNA produced in the cells (Supplementary Fig. 9C, D, E). We observed about one-tenth the concentration of mRNA in cell lysate, compared with plasmid DNA introduced into the cells, scaling linearly with 10−300 nM DNA template.

We further established the generality of the RNA lanterns and tags for imaging model transcripts. We first evaluated an inducible expression system to monitor the production of a fluorescent protein transcript tagged with M-5-P (Supplementary Fig. 10A)[34]. Luminescence turn-on was observed in the first 10 min of induction, with fluorescent signal (following subsequent protein translation and chromophore maturation) appearing later (Supplementary Fig. 10B, C). We also used the lanterns to dynamically trace biologically relevant targets, including β-actin RNA (Supplementary Fig. 11A)[35]. This transcript is known to localize to stress granules upon arsenite

treatment. The lantern and tag set shown in Fig. 5 were used to monitor the RNA over time (Supplementary Movies 1, 2). Punctate-like structures were observed in some cells (Fig. 5D, Supplementary Fig. 11B, Supplementary Fig. 12)[35–38]. However, definitive granule localization could not be confirmed in any case. This is perhaps due to the longer acquisition times required for bioluminescence, complicating the assignment of subcellular features in relation to fluorescent markers (G3BP1, in this case). Similar analyses were performed with another model transcript, *CDK6* (Supplementary Fig. 13, A and B, Supplementary Movies 3–5)[39]. Collectively, these studies illustrate the utility of the structured bait for RNA lantern assembly and transcript visualization, but also highlight some of the inherent limitations with imaging bioluminescent probes.

Finally, as a proof of concept, we imaged RNAs in subcutaneous models in vivo. HEK293T cells that were engineered to express RNA lanterns and mRNAs encoding either GFP or BFP with variable 3' UTRs: M-3-P, M-3-P^mut, or no bait were used. The cells were incubated with luciferin, implanted in *Ai9* mice dorsal flanks, and imaged. Increased luminescence was observed from cells expressing RNA lanterns and

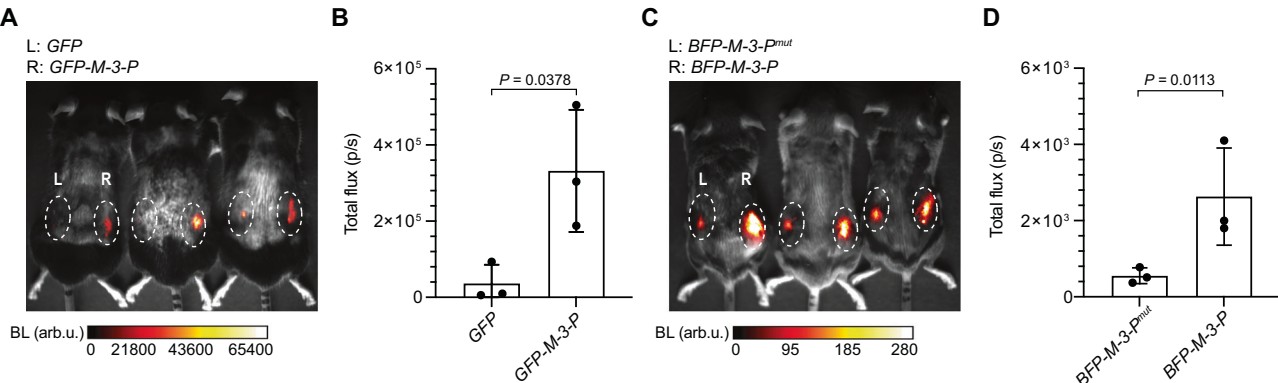

**Fig. 6 | Imaging in live mice with RNA lanterns. A** *Ai9* mice implanted with cells expressing RNA lanterns and *GFP* alone (left flank) or *GFP–M-3-P* (right flank) and imaged with luciferin. **B** Photon flux from *Ai9* mice with implanted cells expressing *GFP(±)M-3-P* and RNA lanterns shown in (**A**). **C** *Ai9* mice implanted with cells expressing RNA lanterns and *BFP–M-3-P^{mut}* (left flank) or *BFP–M-3-P* (right flank).

**D** Photon flux from *Ai9* mice with implanted cells expressing either *BFP–M-3-P* or *BFP–M-3-P^{mut}* and RNA lanterns shown in (**C**). Data are presented as mean values ± SD for $n = 3$ replicates (Fig. 6B, D). $P$ values (95% confidence interval) were determined by unpaired, two-tailed $t$-test. BL bioluminescence. arb.u. arbitrary units. Source data for (**A**–**D**) are provided as Source Data files.

*GFP-M-3-P* (right flank) compared to cells lacking the M-3-P bait (left flank (Fig. 6A, B). Similar increases in luminescence were observed from cells expressing *BFP-M-3-P*, compared to control transcripts in the presence of the lantern components (Fig. 6C, D). These data demonstrate that lanterns have the potential for RNA-dependent imaging in tissue, provided that sufficient levels of the assembled luciferase are present and that sufficient quantities of the luciferin can be delivered. In such cases, the stage is set for real-time detection of gene expression in live animals.

## Discussion
RNA dynamics have been historically visualized in living systems with fluorescent probes. Such methods require optically permissive platforms such as transparent model organisms or surgically implanted windows. The required excitation light can further induce high background signals in tissue and is often limited in depth. Bioluminescent probes (luciferases) overcome some of these limitations. Luciferase reporters use enzymatic reactions—instead of excitation light—to generate photons, which produce far less background signal and can be advantageous for serial imaging in tissue and whole animals.

To capitalize on the sensitivity and dynamic range of bioluminescence, we developed a genetically encoded split luciferase-based platform for visualizing RNA transcripts. The RNA lanterns combine the well-known NanoBiT system and MS2/PP7 platform for transcript tracking. We systematically optimized the components to achieve sensitive imaging. Substantial improvements in signal production were achieved by modulating the spacing and phase angle of the aptamer components. The optimized design significantly decreases the size of previously reported RNA-protein complexes for imaging. Only a single copy of the RNA bait was necessary for imaging target transcripts in cells and whole organisms.

The RNA bait and lanterns also comprise unique features for a range of applications. Both the proteins and bait are equipped with affinity tags for retrieval and downstream analyses of target transcripts and interacting biomolecules. Structured RNA baits can productively assemble diverse split proteins, including Fluc and various BRET reporters. Such modularity expands the color palette of RNA lanterns and sets the stage for even deeper tissue imaging and multiplexed assays. Given that the BRET probes were more efficiently assembled with longer RNA baits, though, care should be taken to optimize each lantern and tag combination.

Our work further highlights that the bioluminescent readout is impacted by several parameters, all of which must be considered in a dynamic imaging study. Signal production is dependent on the concentration of lantern and bait pair, luciferin availability, and the lifetime of the tagged transcript. Lantern levels should be empirically tuned to ensure maximal signal, as detection is diminished when RNA levels are too low or too high (via the hook effect). A range of substrate concentrations and delivery methods should also be examined, because long-lived RNAs will require a continual supply of luciferin for sustained signal production. Finally, the impacts of the tag and bound lantern must be examined on a given transcript target. For example, MS2 is known to stabilize RNAs and may perturb the lifetime of its binding target in cells.

We anticipate that RNA lanterns will enable RNA dynamics to be visualized in vitro and in vivo. The probes will be particularly useful for serial imaging of transcripts, where localization and expression are difficult to examine over time. While we focused on visualizing model transcripts with RNA bait incorporated into the 3′ UTR, the short tag can easily be applied to other transcripts through genetic manipulation. Further tuning of both the RNA lantern and tag will expand the number of transcripts that can be visualized in tandem. Such multiplexed studies will paint a more complete picture of RNA biology in living systems.

## Methods
### Ethical statement
This study was approved by the Institutional Animal Care and Use Committee (IACUC) of the University of California Irvine in compliance with the National Institutes of Health guidelines, under protocol number AUP-21-137.

### General information
Q5 DNA polymerase, restriction enzymes, and all buffers were purchased from New England Biolabs. dNTPs were purchased from Thermo Fisher Scientific. Luria-Bertani medium (LB) was purchased from Genesee Scientific. All plasmids and primer stocks were stored at −20 °C unless otherwise noted. Primers were purchased from Integrated DNA Technologies and plasmids were sequenced by Azenta Life Sciences. Sequencing traces were analyzed using Benchling. All primers used in the studies are provided in Supplementary Data 1.

### General cloning methods
Polymerase chain reaction (PCR) was used to prepare genes of interest, and products were analyzed by gel electrophoresis. Products were excised and purified. Amplified genes were ligated into destination vectors via Gibson assembly[40]. All PCR reactions were performed in a BioRad C3000 thermocycler using the following conditions: 1) 95 °C for 3 min, 2) 95 °C for 30 s, 3) −1.2 °C per cycle starting at 72 °C for 30 s,

4) 72 °C for 30 s, repeat steps 2–4 ten times, 5) 95 °C for 3 min, 6) 95 °C for 30 s, 7) 60 °C for 30 s, 8) 72 °C for 2 min repeat steps 6–8 twenty times, then 72 °C for 5 min, and hold at 12 °C until retrieval from the thermocycler. Gibson assembly conditions were: 50 °C for 60 min and hold at 12 °C until retrieval from the thermocycler. Ligated plasmids were transformed into TOP10 *E. coli* cells using the heat shock method. After incubation at 37 °C for 18–24 h, colonies were picked and expanded overnight in 5 mL LB broth supplemented with ampicillin (100 μg/mL) or kanamycin (100 μg/mL). DNA was extracted from colonies using a Zymo Research Plasmid Mini-prep Kit. DNA was subjected to restriction enzyme digestion to confirm gene insertion. Positive hits were further sequenced. Additional construct details are provided in Supplementary Table 1 and Supplementary Note 1.

### General cell culture methods

HEK293T cells (HEK, ATCC) and stable cell lines derived from HEK293T cells were cultured in complete media: DMEM (Corning) containing 10% (v/v) fetal bovine serum (FBS, Life Technologies), penicillin (100 U/mL), and streptomycin (100 μg/mL, Gibco). Cell lines stably expressing RNA lantern and linker designs were generated via lentiviral transduction. Transduced cells were further cultured with puromycin (20 μg/mL) to preserve gene incorporation. Cells were incubated at 37 °C in a 5% $CO_2$ humidified chamber. Cells were serially passaged using trypsin (0.25 % in HBSS, Gibco).

### Protein expression and purification

LgBiT was encoded in a pCold vector. LgBiT was expressed in *E. coli* BL21 cells grown in LB medium (1 L). Expression was induced at an optical density (OD600) of ~0.6 by addition of 0.5 mM isopropyl-β-D-thiogalactopyranoside (IPTG), followed by incubation at 37 °C for 4 h. Cells were harvested by centrifugation (4000 × *g*, 10 min, 4 °C). Cells were then resuspended in lysis buffer (30 mL, 50 mM Tris HCl, 150 mM NaCl, 0.5% Tween-20, 1 mM phenylmethylsulfonyl fluoride [PMSF], pH 7.4). Cells were sonicated (Qsonica) at 40% amplitude, at 2 s on 2 s off intervals for 15 min. Cell debris was removed through centrifugation (10,000 × *g*, 1 h, 4 °C). Proteins were purified by Ni-NTA affinity chromatography. The columns were washed with wash buffer (20 mM imidazole, 50 mM $NaPO_4$, pH 7.4), and captured proteins were eluted with elution buffer (200 mM imidazole, 50 mM $NaPO_4$, pH 7.4). Samples were dialyzed overnight at 4 °C into phosphate buffer (50 mM $NaPO_4$, pH 7.4), then concentrated to ~500 μL using Amicon Ultra-15 Centrifugal Filter Units (Merck Millipore MWCO 3 kDa). Protein concentrations were determined via absorption measurements (JASCO V730 UV-vis spectrophotometer, 280 nm). SDS-PAGE analyses were also performed to verify purity, and gels were stained with Coomassie R-250.

### In vitro transcription

RNA was transcribed in vitro in a buffer containing 50 mM Tris-HCl pH 8, 2 mM spermidine, 0.01% Triton X-100, 2 mM rNTPs (each), 20 mM $MgCl_2$, 10 mM dithiothreitol (DTT), and recombinant T7 RNA polymerase. Transcription reactions were quenched with 30 mM EDTA and subsequently ethanol-precipitated with coprecipitant (Invitrogen GlycoBlue, AM9516) prior to purification via denaturing PAGE. Purified RNAs were eluted from gel pieces into 300 mM KCl and 0.1 mM EDTA for 3 h. Eluted RNAs were ethanol-precipitated with coprecipitant, and pellets were washed with 70% ethanol, dried, and resuspended in 10 mM Tris-HCl pH 7.5, 0.1 mM EDTA, and 0.001% Triton X-100 (TET). RNA concentrations were determined by UV–vis absorption spectroscopy (Thermo Scientific NanoDrop 2000, ND-2000). All primers used in these studies are provided in Supplementary Data 1.

### In vitro transcription/translation (IVTT) in rabbit reticulocyte lysate

A rabbit reticulocyte lysate (RRL) in vitro translation kit (Promega, L4960) was coupled to in vitro transcription with the supplementation of $MgCl_2$, rNTPs, and T7 RNA polymerase. Plasmid DNAs were 3′ linearized for run-off transcription. RNA scaffolds were prepared via primer elongation of synthetic DNA oligos (Integrated DNA Technologies). Linearized plasmid DNAs and RNA bait DNAs were kit purified prior to IVTT (DNA Clean and Concentrator; Zymo Research, D4004). All DNAs were eluted into Tris-EDTA (TE; 10 mM Tris-HCl and 0.1 mM EDTA, pH 7.5) and controls were supplemented with an equal volume of TE buffer.

IVTT reactions were prepared on ice with 70% v/v lysate, 0.02 mM amino acid mix, rNTPs (0.35 mM GTP and 0.15 mM each ATP/CTP/UTP), 1 mM $MgCl_2$, linearized plasmid DNA, RNA bait template DNA, 2 u/10 μL RNase inhibitor (Invitrogen, AM2694), and 0.5 μL/10 μL reaction of recombinant T7 RNA polymerase (house preparation). NanoBiT experiments were set up with 10 μL volumes and 1 nM of linearized plasmid DNA. Split firefly luciferase experiments were performed with 25 μL volumes and 3 nM of linearized plasmid DNA. RNA bait DNA templates were varied at specified ratios. IVTT reactions were incubated at 34 °C for 10 min and 30 °C for 140 min.

Luciferin substrates were supplied prior to imaging. For split NanoBiT RNA lantern experiments, furimazine was added to each reaction at 20 μM, and samples were incubated at room temperature for 5 min before imaging. For split firefly luciferase RNA lantern experiments, D-luciferin (100 μM) and ATP (1 mM) were added to each reaction. Samples were then incubated at room temperature for 5 min before image acquisition. Plates were imaged in a dark, light-proof chamber using an IVIS Lumina (PerkinElmer) CCD camera chilled to −90 °C. The stage was kept at 37 °C during imaging and the camera was controlled using Living Image software. Exposure times were set to 1 min and binning levels were set to medium. Regions of interest were selected for quantification and total flux values were analyzed using Living Image software. All data were exported to Microsoft Excel or Prism 10 (GraphPad) for further analysis.

### Reconstitution kinetics

Three reticulocyte lysate IVTT reactions (10 μL) were set up as above. Reactions were incubated at 34 °C for 10 min and 30 °C for 140 min. Furimazine (20 μM) was added to each reaction, and then samples were placed in a black, clear bottom 384-well plate. Purified M-3-P RNA (1 nM), M-3-P DNA (1 nM), or *GFP-M-3-P* DNA (1 nM) were added individually to each reaction. The reactions were immediately imaged using an Andor iXon Ultra 888 EMCCD camera equipped with an F0.95 lens, with an EM gain of 1000. Images were acquired (30 s acquisition) over 60 min. Images were processed with ImageJ (Fiji 3) and densitometry was used to measure mean intensity values over time. All data was exported to Microsoft Excel for analysis and then plotted using GraphPad Prism 10.

### In cellulo linker optimization

Stable cell lines expressing variant RNA lanterns were plated in clear 12-well plates and incubated overnight. Cell lines were then transfected with 1 μL Lipofectamine 3000 (Invitrogen, L3000001), 2 μL P3000 Reagent (Invitrogen, L3000001), and 500 ng of the BFP-M-3-P plasmid. After 24 h incubation, cells were lifted and counted with a Countess II (Invitrogen). 50,000 transfected or non-transfected cells were plated in triplicate into black 96-well plates (Greiner Bio-One). Furimazine (20 μM) was added to each sample. Plates were imaged in a dark, light-proof chamber using an IVIS Lumina (PerkinElmer) CCD camera chilled to −90 °C. The stage was kept at 37 °C during imaging and the camera was controlled using Living Image software. Exposure times were set to 1 min and binning levels were set to medium. Regions of interest were selected for quantification and total flux values were analyzed using Living Image software. All data were exported to Microsoft Excel or Prism 10 (GraphPad) for further analysis. The remaining cells were analyzed on a Novocyte 3000 flow cytometer (ACEA BioSciences) for BFP expression.

## General flow cytometry methods

Cells were treated with trypsin for 5 min at 37 °C and then neutralized with complete media. Cells were transferred to Eppendorf tubes and pelleted ($500 \times g$, 5 min) using a tabletop centrifuge (Thermo Fisher Sorvall Legend Micro 17). The resulting supernatants were discarded, and cells were washed with PBS ($2 \times 400\,\mu L$). Cells were analyzed for XFP expression on a Novocyte 3000 flow cytometer. Live cells were gated using FSC/SSC settings, and singlet cells were further gated (Supplementary Fig. 14). For each sample, 10,000 events were collected on the "singlet cell" gate.

## HEK293T lysate preparation

HEK293T cells stably expressing RNA lanterns (~$10^6$) were pelleted at $500 \times g$ for 10 min at 4 °C and washed three times with binding buffer (140 mM KCl, 10 mM NaCl, 20 mM Tris-HCl pH 7.5, and 5 mM $MgCl_2$). Cells were pelleted during each washing step at $500 \times g$ (1 min, 4 °C) After the final wash, the cell pellet was resuspended in 600 μL binding buffer supplemented with 1% Tween-80, 1X protease inhibitor cocktail (Roche, 11697498001), and 5 μg RNase inhibitor. Cells were sonicated on ice (5 cycles of 10 s on, 10 s off; Branson CPX2800). Cell debris was then removed by centrifugation ($12,000 \times g$, 4 °C, 15 min). The resulting supernatant was aliquoted and stored at −80 °C for future use as 2X lysate stocks.

## RNA titration assay

Purified RNAs (10X stocks in TET buffer) were serially diluted. Binding reactions contained 12.5 μL of 2X cell lysate, 10 μL of 1X binding buffer, and 2.5 μL of 10X RNA. Reactions were incubated at room temperature on an orbital shaker for 30 min. After incubation, 1 μL of a 200 μM furimazine stock (Promega, Nano-Glo substrate) in 1X binding buffer was added to each reaction and then plated in a black, clear bottom 384-well plate and imaged with an Andor iXon Ultra 888 EMCCD camera (Oxford Instruments) equipped with an F 0.95 lens (Schneider) with an EM gain of 500. Images were acquired (5 s acquisition) over 10 min. Images were Z-stacked in ImageJ (Fiji 3) and densitometry was used to measure peak intensity values.

## Ni-NTA pull-down

Triplicate reactions of selected RNA concentrations above, within, and below the curve (as determined by RNA titration assay) were combined (45 μL/each). Charged Ni-NTA agarose beads (~45 μL; Thermo Fisher, 88221) were washed three times in 90 μL 1X binding buffer in a spin-column (Corning, 8160) at $3000 \times g$ for 30 s. The combined binding reactions were then incubated on the Ni-NTA agarose at room temperature for 30 min on an orbital shaker. After incubation, the columns were spun at $3000 \times g$ for 30 s and the flowthrough was collected. Four 5-min washes were performed with 45 μL 1X binding buffer at room temperature on an orbital shaker. After each incubation, the columns were spun, and the flowthroughs were collected (washes 1–4). Columns were eluted three times with 1X binding buffer supplemented with 25 mM EDTA (20 mM final) and incubated for 5 min on an orbital shaker. Elution fractions were collected. Furimazine was added to the fractions (20 μM final) and the solutions were placed in a black, clear bottom 384-well plate. Images were acquired using an Andor iXon Ultra 888 EMCCD camera equipped with an F 0.95 lens, with an EM gain of 500. Images were acquired (10 s acquisitions) over 15 min. Final images were Z-stacked, and densitometry (ImageJ Fiji3) was used to determine fold change over background and no RNA controls. Final images were overlaid with brightfield photos. Plots were produced using GraphPad Prism 10.

## HA and FLAG pull-down

Reticulocyte lysate IVTT reactions (67.5 μL) were set up as described in the In vitro transcription/translation (IVTT) in rabbit reticulocyte lysate methods section. To each reaction, 100 nM M-3-P RNA (7.5 μL) or 1X binding buffer (7.5 μL) was added prior to incubation. Reactions were incubated at 34 °C for 10 min and 30 °C for 140 min. Pierce magnetic anti-HA beads (45 μL, ThermoFisher) and anti-FLAG beads (1.5 μL, ThermoFisher) were normalized for binding capacity. Two sets of each bead were washed three times in 1X binding buffer ten-times their initial volume (450 μL and 15 μL, respectively; 5 min with agitation). To each tube, 75 μL of IVTT reactions were added and incubated with agitation for 60 min at room temperature. The supernatant was removed and stored separately. After incubation, beads were washed three times in 1X binding buffer (75 μL, 5 min with agitation) and each wash was stored. Beads were resuspended in 30 μL 1X binding buffer and 20 μM furimazine, then plated in triplicate in a black, clear bottom 384-well plate and imaged with an Andor iXon Ultra 888 EMCCD camera equipped with an F 0.95 lens, with an EM gain of 100. Images were acquired (5 s acquisition) over 15 min. Images were Z-stacked in ImageJ (Fiji 3) and densitometry was used to measure peak intensity values.

## Lantern complex isolation and quantification

HEK293T cells ($5 \times 10^5$) expressing RNA lanterns were added to 12-well plates, and then transiently transfected 24 h later with DNA (0–1000 ng) encoding *GFP–M-3-P* (Lipofectamine 3000). Cells were lifted 48-h post-transfection, washed, and counted. Cells ($5 \times 10^4$) were then analyzed for luminescence intensity using a luminometer (Tecan). The remaining cells were then resuspended in ~300 μL of PBS and divided evenly into two separate 1.5 mL centrifuge tubes. Cells were pelleted at $500 \times g$ for 10 min (4 °C), and media was removed. The pellets were stored at −80 °C for RNA analysis.

**Ni-NTA pull-down lysate preparation.** Frozen cell pellets from above were thawed on ice, then resuspended in 100 μL of 1X binding buffer (140 mM KCl, 10 mM NaCl, 20 mM Tris-HCl pH 7.5, and 5 mM $MgCl_2$) supplemented with 1% Tween-80, 1X protease inhibitor cocktail (Roche, 11697498001), 20 U of SUPERase•In™ RNase Inhibitor (Invitrogen), and 20 μg/mL puromycin. Cells were lysed using a 30G syringe needle (10 plunges) and the cell debris was pelleted by centrifugation ($12,000 \times g$, 4 °C, 15 min). The resulting supernatant was discarded and the pellet was resuspended in 100 μL of 1X binding buffer supplemented with 1% Tween-80, 1X protease inhibitor cocktail, and 20 U of SUPERase•In™ RNase Inhibitor. The lysate was clarified by centrifugation ($3000 \times g$, 30 s) on a Spin-X column and stored on ice prior to use.

**Ni-NTA pulldown.** Furimazine was added to the lysate above (20 μM), and samples were placed in a black-bottom, 96-well plate. Images were acquired using an Andor iXon Ultra 888 EMCCD camera equipped with an F0.95 lens, with an EM gain of 1000. Images were acquired (30 s acquisitions) over 5 min. Final images were Z-stacked, and densitometry (ImageJ Fiji3) was used to measure peak intensity values.

HisPur™ Ni-NTA magnetic beads (5 μL) per condition were washed three times with 1X binding buffer supplemented with 0.1% Tween-80 (50 μL; 5 min with agitation). Prior imaged lysate was added to each tube, and samples were incubated with agitation for 5 min at RT. The flow-through was removed and stored separately on ice. Beads were washed three times in 1X binding buffer supplemented with 0.1% Tween-80 (100 μL, 10 s with agitation) and each wash was stored on ice. Beads were resuspended in 1X binding buffer supplemented with 0.1% Tween-80 (100 μL). Furimazine was added to the flow through, washes, and beads (20 μM), and samples were then plated in a black-bottom, 96-well plate. Images were acquired using an Andor iXon Ultra 888 EMCCD camera equipped with an F0.95 lens, with an EM gain of 1000. Images were acquired (30 s acquisitions) over 5 min. Final images were Z-stacked, and densitometry (ImageJ Fiji3) was used to determine the fold change of RNA lanterns captured on the beads over background and no mRNA control.

**Phenol-chloroform extraction.** *GFP-M-3-P* mRNA pulled down on the Ni-NTA beads was transferred to a 1.5 mL centrifuge tube (100 µL). Equal volumes of phenol:chloroform:isoamyl alcohol (25:24:1) were added, and samples were vortexed for 30 s. Samples were then centrifuged at 14,000 × *g* for 3 min, and the aqueous phase was collected and transferred to a new tube. The extraction was repeated an additional two times for a total of three extractions. RNA was ethanol-precipitated overnight at −80 °C with 1 µL GlycoBlue co-precipitant (ThermoFisher) and 100 mM KCl. The pellets were washed twice with cold 70% ethanol, dried, and stored at −80 °C for future use.

**TRIzol extraction.** Frozen HEK293T cell pellets were thawed on ice and resuspended in TRIzol Reagent (500 µL, Invitrogen). The RNA was isolated following the manufacturer's instructions. A portion of the aqueous phase (290 µL) for each condition was then precipitated with isopropanol (500 µL, RT). Samples were pelleted by centrifugation at 12,000 × *g* (20 min, 4 °C), washed twice with cold 70% ethanol, dried, and stored at −80 °C for future use.

**RT-qPCR.** RNA pellets from both the phenol-chloroform and TRIzol extractions were treated with DNase I (New England Biolabs) following the manufacturer's protocol, then column purified using a Zymo Research RNA Clean and Concentrator kit. The RNA was reverse transcribed using a reverse primer for *M-3-P* and *Bst* 3.0 DNA polymerase (New England Biolabs)/ProtoScript II reverse transcriptase (New England Biolabs). Quantitative RT-PCR was performed on a Bio-Rad CFX Connect system using Luna® Universal qPCR Master Mix (New England Biolabs). Designed primers were acquired from Integrated DNA Technologies and are provided in Supplementary Note 1. The initial DNA quantity of each sample was determined by interpolation of the quantification cycle (Cq) from a standard curve using the *GFP-M-3-P* DNA and primer set.

## Bioluminescence microscopy

HEK293T cells ($5 \times 10^5$) stably expressing RNA lanterns were plated in 8-well Ibidi µ-Slides. After 24 h, the cells were transiently transfected with 100 ng of *GFP-M-3-P* plasmid or GFP plasmid using 0.3 µL Lipofectamine 3000 and 0.3 µL P3000 reagent. After 18 h, cell media was exchanged for phenol red-free DMEM (Gibco FluoroBrite DMEM, A1896701) supplemented with 20 µM furimazine. Live cell static images were captured on an Olympus IX71 microscope equipped with an Andor iXon Ultra 888 EMCCD camera and a 40× oil objective (olympus UPlanApo 40×/1.00 oil iris). For live cell static images with a larger field of view, a 4× objective (Olympus PlanC N 4×/0.10 na) and 10× objective (Olympus UPlanFL N 10×/0.30 Ph1) were used. Images were captured with an EM gain of 1000, 10 MHz horizontal read-out rate, 4.33 µs vertical clock speed, and an acquisition time of 180 s. The microscope stage was kept warm with a heating pad to maintain cell viability and to encourage enzyme turnover. Fluorescence images were captured immediately following luminescence imaging, using a blue LED light source (ThorLabs, Solis-470C). Fluorescent images were captured using the EMCCD's conventional mode with an acquisition time of 0.5 s. Images were processed (removed outliers and Z-stacked) with ImageJ (Fiji 3) and colocalization was determined with the JaCoP plug-in[41].

## Dynamic imaging of cellular stress

HEK293T cells ($5 \times 10^4$) stably expressing RNA lanterns were plated in eight-well Ibidi µ-Slides coated with (10 mg/cm$^2$) fibronectin. After 24 h, the cells were transiently transfected with 200 ng of plasmid encoding *mCherry-β-actin* with M-3-P in the 3′ UTR or *CDK6-M-3-P-IRES-Staygold*. Some cells were also transfected with 200 ng of plasmid encoding *GFP-G3BP1* (Addgene #135997). Transfections were performed using Lipofectamine 3000 (Invitrogen, 0.4 µL) and P3000 Reagent (Invitrogen 0.4 µL). After 18 h, cell media was exchanged for

phenol red-free DMEM (Gibco FluoroBrite DMEM, A1896701) supplemented with 30 µM furimazine and 0.5 mM sodium (meta)arsenite. Live cell images were captured on an Olympus IX71 microscope equipped with a HNu 512 EMCCD camera and a 40× oil objective (Olympus UPlanApo 40×/1.00 oil iris). Luminescence images were captured with an EM gain of 1000, 10 MHz horizontal read-out rate, 7.55 fps, and an acquisition time of 90 s. Fluorescence images were captured immediately before and following luminescence imaging, using either a blue or green light source (Lycco Flashlight). Fluorescence emission was filtered using GFP (Olympus UM52) and Cy3 (Chroma UN41001 FITC CIN43662) filters. Fluorescence images were captured using an EM gain of 1000 with an acquisition time of 1 or 2 s. Images were processed (removed outliers and Z-stacked) with (Fiji 3) and colocalization was determined with the JaCoP plug-in[41]. Where needed, images were aligned using the Linear Stack Alignment with SIFT plugin[42].

## Immunohistochemistry

Cells were washed with PBS ($3 \times 100$ µL) and fixed using 4% PFA (RT, 10 min). The samples were then washed with PBS containing 0.025% v/v Triton X-100 (PBS-T, $3 \times 100$ µL) and permeabilized with 0.1% v/v Triton X-100 (RT, 15 min). Following permeabilization, samples were washed with PBS-T ($3 \times 100$ µL) and incubated with rabbit α-G3BP1 (E9G1M) XP® (Cell Signaling Technology) in blocking buffer (1% BSA, 0.01% NaN$_3$, 1:100) or blocking buffer only overnight. The next day, the samples were washed with PBS-T ($3 \times 100$ µL) and stained with an Alexa Fluor 594 goat α-rabbit IgG secondary antibody (Invitrogen) in blocking buffer (1:200, 1 h, RT). Samples were washed with PBS ($3 \times 100$ µL, 5 min per wash) and then imaged on a Keyence (BZ-X800) microscope with a 40× objective (Keyence Plan Apochromat 40×). Fluorescence was captured using Cy5 (BZ-X Filter Cy5) and GFP (BZ-X Filter GFP) filter cubes for G3BP1 and Staygold expression, respectively.

## Imaging of doxycycline-inducible RNA bait

HEK293T cells ($5 \times 10^4$) stably expressing RNA lanterns were cultured in doxycycline-free media. Cells were plated in 8-well Ibidi µ-Slides coated with fibronectin (10 mg/cm$^2$). After 24 h, the cells were transiently transfected with 200 ng *TRE2-XFP-M-5-P* using Lipofectamine 3000 (Invitrogen) according to the manufacturer's instructions. After 18 h, cell media was exchanged for phenol red-free DMEM (Gibco FluoroBrite DMEM, A1896701) supplemented with 30 µM furimazine and 5.12 µg/mL doxycycline hyclate (TCI Chemicals). Live cell images were captured on an Olympus IX71 microscope equipped with a HNu 512 EMCCD camera and a 20× objective (Olympus LUCPlanFL 20X Ph1). Luminescence images were acquired using an EM gain of 1000, 10 MHz horizontal read-out rate, 7.55 fps, and an acquisition time of 90 s. Fluorescence images were acquired immediately before and following luminescence imaging, using a green light source (Lycco Flashlight). Fluorescence emission was filtered by a Texas Red (Olympus UM52) filter. Fluorescence images were captured using an EM gain of 1000 with an acquisition time of 4 s. Images were processed (removed outliers and Z-stacked) with ImageJ (Fiji 3) and colocalization was determined with the JaCoP plug-in[41].

## In vivo cell transplants

The strain used in this study was B6;129S6-Gt(ROSA)26Sor$^{tm9(CAG-tdTomato)Hze}$/J commonly known as Ai9. The species was *Mus musculus* obtained from the Jackson Laboratory (JAX: 007905). All mice used were male, there was no preference for sex, the selection was made based on the availability of animals in the colony. Mice were maintained on a 12 h light/dark cycle at 25 °C with humidity kept between 30% and 70%. All procedures were done during the light portion of the cycle.

A total of six mice were used: three were 12 weeks old, and the remaining three were 9 weeks old at the time of injection. The mice were anesthetized using isoflurane (1–2%) and were given

subcutaneous dorsal injections of HEK293T cells ($1 \times 10^6$) expressing RNA lanterns and *GFP*, *GFP-M-3-P*, *BFP-M-3-P$^{mut}$*, or *BFP-M-3-P* suspended in sterile PBS solution. Cells were normalized for mean fluorescence intensity (MFI) of *XFP* using flow cytometry. Following normalization, cells were implanted into the dorsal posterior subcutaneous flank of each mouse. Cells expressing RNA lantern and controls, *GFP* or *BFP-M-3-P $^{mut}$*, were implanted on the left side, while cells expressing the RNA lantern and *GFP-M-3-P* or *BFP-M-3-P* were implanted on the right side, allowing each mouse to serve as its own control. Each animal received a 20 µM furimazine injection in 200 µL of the implant.

Animals were imaged immediately following transplantation. For imaging, animals were anesthetized with an i.p. injection of ketamine (6.6 mg/mL) and xylazine (1.65 mg/mL) suspended in sterile PBS and placed on a warmed (37 °C) stage. Images were captured on an Andor iXon Ultra 888 EMCCD camera equipped with an F 0.95 lens. Images were acquired with an EM gain of 1000, an acquisition time of 300 s, and signal was captured as photons. Images were processed (removing outliers and integrated density) using ImageJ (Fiji 3). Total flux (p/s) was determined based on a given region of interest (ROI) for each implantation. Background correction was accomplished by averaging ROIs containing no luminescence. Prism 10 (GraphPad) was used to determine significant differences (unpaired, two-tailed *t*-test) between groups.

### Statistics and reproducibility

Prism was used for all data analysis and graph plotting. Data are presented as mean values ± SD. The results were analyzed by unpaired two-tailed *t*-test between two groups. Exact *P* values were provided accordingly in the captions. $P < 0.05$ was used as the threshold for statistical significance; (*) indicates $P < 0.05$, (**) indicates $P < 0.01$, and (***) indicates $P < 0.001$. All statistical analyses were performed with Prism 10 and exact *P* values were calculated using the =T.DIST.2 T(t, df) function on Microsoft Excel. No statistical method was used to predetermine the sample size. The exact number of replicates and statistical tests are indicated in the figure legends. Unless otherwise indicated, *n* represents the number of independent experimental replicates.

### Reporting summary

Further information on research design is available in Nature Portfolio Reporting Summary linked to this article.

## Data availability

The authors declare that all relevant data that support the findings of this work are presented in the Article, the Supplementary Information, and the Source Data files. Source data are provided with this paper. Plasmid DNAs and additional data are available from the corresponding authors upon request. Source data are provided with this paper.

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

## Acknowledgements

We thank Dr. Lorenzo Scipioni and Prof. Michelle Digman (UCI) for their help with optical imaging experiments. We also thank members of the Luptak, Steward, and Prescher laboratories for helpful discussions. This work was supported by the W.M. Keck Foundation: (O.S., A.L., J.A.P.); NASA ICAR 80NSSC21K0596 (A.L.); The Paul G. Allen Frontiers Group (J.A.P.); National Science Foundation 1804220 (A.L.); National Institute of Health grant R01CA229696 (C.C.C.).

## Author contributions

C.C.C., O.S., A.L., J.A.P. conceived the project idea; L.P.H., K.H.C., K.K.N., A.L., J.A.P. developed the methodology; L.P.H., K.H.C., K.K.N., E.B.F., C.E.T.C., C.C.C., E.R.-M., A.D.K., Z.R.T., A.L., O.S., J.A.P. designed experiments; L.P.H., K.H.C., K.K.N., E.B.F., C.E.T.C., C.C., M.M., C.C.C., A.A.B., E.R.-M., A.D.K. performed experiments; L.P.H., K.K.N., K.H.C., E.B.F. conducted imaging analyses; L.P.H.,. K.K.N., K.H.C., A.L., J.A.P. wrote the initial manuscript draft. All authors participated in manuscript reviewing and editing.

## Competing interests

A.L., J.A.P., O.S., K.K.N., K.H.C., L.P.H., and C.C.C. are on a provisional patent application, filed through UCI, based on the results described here. The remaining authors declare no competing interests.
