## [Transparent Peer Review file · Nature Communications]

A modular platform for bioluminescent RNA tracking

Corresponding Author: Professor Jennifer Prescher

Version 0:

Reviewer comments:

Reviewer #1

(Remarks to the Author)

The authors describe the development of a bioluminescence-based RNA visualization method using split NLuc reconstitution and RNA recognition by MCP and PCP. A target RNA is fused with a tag RNA region, including MS2 and PP7, which are selectively captured by MCP and PCP, respectively. The MCP and PCP are fused with split NLuc fragments of SmBiT and LgBiT, respectively. Upon the MCP-SmBiT and PCP-LgBiT binding to the tag RNA's MS2 and PP7 regions, the fragments of NanoBiT bring closely to come together and reconstitute the split NLuc. The authors also tried to replace NLuc with Fluc and to adopt the BRET technique, allowing variation of ranges of photon flux amounts and emission wavelengths. Finally, the method was applied to RNA imaging in cells and in vivo. This reviewer felt that the experiments are carefully designed, and in principle has many advantages over existing methods. However, all the characterization experiments were conducted in in vitro translation system (IVTT), and there were no information on the detection limit, dynamic range, or quantitiveness. Thus, while individual experiments and system optimization seem to be conducted appropriately, characterizations and applications were not sufficiently conducted to demonstrate the practical performance of the present system. In addition, several concerns listed below are raised that the authors need to be addressed.

1. In RNA tracking, the bait would be used as a tag fused to the target RNA to detect and visualize the RNA. The authors should show more applicability to fuse the bait to different RNAs that are meaningful biologically to detect or visualize.
2. The title of this manuscript says "tracking," but there were no experiments of tracking that required time-course analysis.
3. The performance of the RNA detection and visualization system can be expressed by such parameters as the detection range and limit of the target RNA concentration, which are calculated based on the concentration of the bait RNA and the lantern proteins. However, the characterization experiments were performed only in IVTT, and no RNA bait and lantern concentration estimations were conducted. Therefore, the performance of this system cannot be quantitatively evaluated.
4. All the bar graphs other than those in Fig. 4 are accompanied by error bars that depict the standard error of the mean, but the data points in each bar graph have large deviations. As the deviations in bioluminescence intensities would often be inevitable, the degree of the deviations should be evaluated by showing standard deviations to evaluate the quantitative capability of this method. Especially the data in Fig. 3G highly deviate, and it is important to show that there are significant differences to conclude that M-5-P had the strongest luminescence among the RNA baits.
5. Although noticed in the brackets, the expression of three types of SmBiT (high/99, low/114, and ultra/86) needs to be consistent between the main text and the legend of Fig. S1.
6. In Fig. S1B, many potential readers of this article may not be familiar with the word "Retic". The reviewer recommends revising it to reticulocyte, as written in the main text.
7. The bars of SmBiT low and 3x-SmBiT low in Fig. S1B seem identical. It is, of course, OK if the values are accidentally quite similar, but the authors should check for the data mix-ups.

Reviewer #2

(Remarks to the Author)

This manuscript describes the development of a bioluminescent RNA tracking system. The system relies on the well-studied MS2/MCP and PP7/PCP systems in which an RNA stem loop recruits a cognate binding protein (MCP or PCP). This system has traditionally been used for fluorescence detection of RNA by fusing a fluorescent protein, split fluorescent protein, or HaloTag to the MCP or PCP. Here, the authors expand the utility of this system with two significant advances: 1) they fuse components of NanoBiT (smBiT and lgBiT) to MCP and PP7 and optimize the fusions, and 2) they combine the MS2 and PP7 stem loops into a single optimized RNA bait. The authors demonstrate that a single RNA bait gives rise to

bioluminescence signal above background in vitro, in cells, and in vivo. This is an exciting development and an important addition to the RNA imaging toolbox. But the field of RNA imaging is advancing quickly. Many flashy preliminary reports turn out to be poor tools (e.g. performance of peppers and mango in original reports versus in Bühler et al, Nature Chem Bio, 2023). Therefore, while the RNA lantern tool represents an exciting advance, there are some important benchmarking experiments that should be done to provide the field with a more comprehensive analysis of this exciting new tool. Upon completion of these experiments, I would be supportive of publication in Nature communications.

1) Characterization of RNA detection in cells is too preliminary and needs to be expanded. In Fig 4B, 4 cells are visible in the GFP channel, whereas only 2 cells are visible in the luminescence channel (the same is true in images in Fig S7); this seems problematic and suggests the correlation between RNA expression (as defined by GFP signal) and bioluminescence signal needs to be defined much more rigorously (The Pearson's coefficient is a population average and not terribly informative). It is not clear what each data point in 4C represents (1 cell? 1 field of view? The average of a field of view from one experiment?) but the standard for imaging studies is that measurements need to be made on tens to hundreds of individual cells. Given the apparent variability in 4B and S7, the authors should not only report the total bioluminescent signal in individual cells for cells expressing the bait versus those expressing a non-binding transfected RNA (like GFP). But they should also report what fraction of cells that show a GFP signal above background also show a bioluminescent signal (for both the bait-expressing and control cells). Perhaps a useful way to do this would be to report the GFP intensity versus BL intensity for individual cells.

2) The structured RNA bait is a clever idea and a valuable contribution to the field. However, for many users the goal of RNA imaging is to study RNA biology. It seems important to provide some basic measurements on whether the structured bait (and the binding of the lantern to the bait) alters RNA localization or stability. For example, it has been shown that the MS2 stem loop interaction with MCP stabilizes RNA transcripts and hence alters the half-life of RNA decay. This could be done by smFISH (for localization) or RT-qPCR for half-life. Similarly, it seems important that the authors characterize the relationship between RNA decay and BL signal decay. What happens when the tagged RNA decays? Does NanoBit remain reconstituted and competent to emit photons? As noted by the authors, there is intense interest in tracing the lifecycle of RNAs in mammalian cells and animals. This requires a tool that emits a signal when the RNA appears and stops emitting the signal when the RNA decays.

3) What are the kinetics of reconstitution? The IVTT seems like it would be the perfect system to define this because you could look at the appearance of luminescence signal after production of the M-3-P RNA bait. It seems this would be important to define because some reconstitution systems can be slow (for example split-GFP 1-10/11 can take 30-60 min for fluorescence signal reconstitutions). This would be important for potential users to evaluate when after export to the cytoplasm and RNA of interest would be visible.

4) The data in 2C are very important/valuable to define the detection sensitivity. The fold change in BL signal over no RNA control is 20x, whereas in vitro the turn on is 300x. This is not surprising as most fluorogenic aptamers exhibit a diminution in signal-to-background in cells compared to in vitro. But it would be valuable for the authors to explicitly discuss the decrease and speculate on the reasons. Is this related to the expression level of the lantern in cells. Since each component of the lantern has an affinity tag, is it possible for the authors to quantify how much of the lantern is expressed in their stable cell line?

5) For the in vivo proof of concept experiment: why were the cells incubated with luciferin prior to implantation? It seems unlikely that users interested in tracking RNA in vivo would follow this experimental paradigm. Why not implant the cells, then deliver luciferin to the mouse? The experimental design seems particularly contrived.

6) For the BRET experiments in Fig 3G: Isn't the fluorescent output what matters here with respect to signal turn on (reconstitution of NanoBit leads to energy transfer to the fluorescent protein)? What is the yellow signal and red signal above the M-3-Pmut or no RNA control?

Minor comments:

- Fig 3B – Here the M-3-P only shows ~80x BL signal over no RNA control whereas in 1E it shows 300x? What is the reason for the variability?
- The data in Fig 2C are valuable for defining detection sensitivity. It would be valuable for potential users to put these RNA concentrations in context. What is the typical expression of a low, medium, highly expressed RNA?
- Page 7, please describe what you mean by “hook effect”

Reviewer #3

(Remarks to the Author)

J. Prescher and colleagues have developed a genetically encoded split luciferase system to visualize RNA transcripts, utilizing the NanoBit system and MS2/PP7 platform. To enhance sensitivity, they optimized components by adjusting the spacing and phase angle of the MS2/PP7 platform. The platform, requiring only a single copy of RNA bait for imaging in cells and organisms, incorporates features for diverse applications, including affinity tags for retrieval and downstream analyses. While another group has recently explored the use of split luciferase to visualise targeted RNA, but with poor performance, J. Prescher and his co-authors have developed an alternative system with much better performance. The design is ingenious, albeit intricate. It offers the potential to explore a broad spectrum of transcripts, providing a promising route for enhancing our understanding of RNA biology in vivo.

While the manuscript is well-written and the authors' conclusions are generally supported by the data, some major and minor points need addressing before the paper is suitable for publication in Nature Communications.

Major points

The literature shows a growing interest in the challenging visualization of transcripts in their real environment. It is difficult to assess the performance of this new system without a suitable benchmark. It would be useful to compare this new system

with the unsplit NanoBiT system. This control is missing.

For real-time detection, it is important to determine the signal intensity as a function of time in order to know the time window for use. Time-dependent bioluminescence intensity experiments are missing. What is the duration of the signal in relation to unsplit NanoBiT?

The notion of small size is relative. The TAG used is 69 nucleotides long, i.e. approximately 23 kDa, which could pose problems when introduced in small non-coding RNAs. The term small should be deleted from the abstract. What impact does the tag have on transcript expression and functionality?

Minor points

The bibliography should be updated to include the review by Palmer et al. published in Cell Chemical Biology on September 17, 2020.

Page 7: "were observed with the additional G4S units compared to the original lanterns both in vitro and" This sentence is incomplete

Page 16, last line: The results of at least 3 experiments are needed to calculate a standard deviation for (D).

The way in which the signal intensity is calculated is not specified in the experimental section. For example, it is not clear what bandwidth (filter), acquisition time or equipment is used. These details are missing from the experimental section. The structure of luciferins and their maximum emission should be given in SI

Version 1:

Reviewer comments:

Reviewer #1

(Remarks to the Author)

The authors have revised the manuscript sufficiently for the most part, adding experimental data in response to the reviewer's comment. However, for some additional data it is unclear which part of the data to focus on to understand the authors' argument. Improvements are needed in the presentation of the data.

In response to comment 1, the authors performed experiments to monitor beta-actin mRNA and CDK6, which were concentrated in the RNA granules when the cell was stressed, as mentioned in the manuscript. However, it is not clear from the microscopic images shown in Figure 5D whether the RNA granule marker (GFP-G3BP1) and luminescence (RNA) are merged well even in the enlarged images 60 min after stress application. The authors should compare the localisation of the RNA granule marker and the luminescence signals before and after stress addition on multiple cells (it is natural that some cells do not form stress granules; overexpression of the granule marker may make it difficult to visualize precisely the RNA granules) and then perform the quantitative analysis of the colocalisation such as Pearson's correlation coefficient. In addition, it may be reader-friendly if the authors add some arrowheads to the images to indicate the colocalised areas.

Reviewer #2

(Remarks to the Author)

This resubmission is an improvement. However there are a number of previous concerns which were not adequately addressed (see below). I remain supportive of publication of this work in Nature communications if the comments outlined below can be addressed. This largely involves editing the manuscript to more clearly articulate the strengths and limitations of the tool and acknowledge important experimental factors that have not yet been defined because they are beyond the scope of the manuscript.

1) I think the concern articulated in my first comment was misinterpreted. I understand that GFP signal was used to identify transfected cells; my concern is that it seems only a subset of the transfected cells show luminescence signal suggesting poor detection of RNA in cells. The manuscript text is misleading on this point and implies that transfected cells show luminescence signal. However, Figure S7 now presents a quantification of the correlation and it is only 62%. I think this needs to be flushed out and addressed directly in the manuscript. It seems important for potential users of this technology to know how well the tool performs at the single cell level. The authors should quantify the correlation (what % of GFP+ cells also show luminescence signal), report this in the main manuscript, and discuss the limitations of the technology that influence this value. For example, in the rebuttal the authors attribute this performance to the "hook effect" and "time-dependent nature of the signal". The hook effect is an important consideration as the performance of the tool (i.e. its ability to detect RNA) diminishes at high and low RNA concentrations. This will directly impact the type of RNA transcript that the tool can effectively detect and the authors should discuss what types of transcripts are likely to be able to be detected by RNA lanterns.

I am not sure what the authors mean by "time-dependent nature of the signal" (the fact that the luminescence decays or that detection sensitivity depends on integration time). Either way, if the time-dependent nature of the signal (which is not a feature of fluorescent reporters) represents a limitation of the tool, again this should be discussed in a way that helps potential users identify appropriate experimental "fits" for this tool. Also, although the authors state they quantify the fraction of bioluminescent cells in Fig S7 and S8, they only present a quantification in S7.

2) Unfortunately, the new data (Figure 5, Supplementary Figure 11 - 13, videos) presented on localization of transcripts to stress granules are not convincing. Either due to low magnification, poor signal to noise, or low resolution, stress granules are not evident in images or movies. The only image that shows something that *could* be construed as stress granules was cell #3 in Figure S13, but unfortunately the strangely punctate signal was present before addition of arsenite, so can't be attributed to stress granules. None of the new data show colocalization of RNA lanterns with G3BP1 in stress granules. Therefore, I disagree with the authors' conclusion that "transcript localization to stress granules was observed". Further, there is no quantification of recruitment/localization (what fraction of stress granules show transcript localization) and there is no negative control where (for example) cells transfected with the mutant M-X-P show no colocalization with stress granules). I understand the challenge in showing stress granules using bioluminescence. In my original comment, I was not suggesting the authors try to track transcript movement at the subcellular level. This is not an inherent strength of bioluminescence. In fact, it is a weakness compared to fluorescence. Rather, bioluminescence is particularly valuable for animal imaging.

In my original comment, I was trying to express concern that the structured aptamer could alter the properties of the RNA transcript to which it is attached, in particular the half-life of the RNA (because the MS2 system has previously been shown to alter mRNA half-life). This could have been addressed by comparing the expression level of XFP mRNA with and without the M-3-P tag. In the revised manuscript, the authors have not addressed whether the M-3-P tag perturbs the transcript to which it is bound. In that case, I would recommend the authors add to the discussion that future experiments should be conducted to determine whether the structured tag perturbs RNA lifetime. As for the stress granule experiments, I don't think the authors' claims or conclusions are supported by the data presented. That said, I don't think they are important for the manuscript (users would be unlikely to use RNA lanterns for subcellular RNA detection) so I would recommend removing them.

In my original question, I asked whether the bioluminescence signal correlated with the presence of RNA in cells (i.e. if the RNA decays does the bioluminescence signal disappear). I understand the authors' rebuttal that the kinetics of the signal is complicated as it depends on RNA lifetime, lantern lifetime, and bioluminescence decay. It would be good for the authors to systematically and rigorously characterize this in future publications. But I would appreciate it if the authors could provide an explanation for the observation in Figure S9B that the majority of the bioluminescence signal doesn't "pull down" with the RNA (i.e. flows through and doesn't bind to the NiNTA column suggesting it isn't associated with the NiNTA aptamer in the tag).

3) The authors have addressed my question with very nice new data presented in Figure 3.

4) The authors addressed my question in the rebuttal but did not sufficiently address it in the manuscript. This could be remedied by straightforward editing of the manuscript. The upshot seems to be that detection depends on RNA concentration, concentration of the protein components, and also probably some complicated kinetics with respect to RNA lifetime and bioluminescence lifetime. None of these caveats are explicitly mentioned in the text. When the authors discuss performance in cells, they should directly address these factors. One thing I don't fully understand about the Supplementary 9 data: in 9B the majority of the luminescence signal comes through in the flow through. Does that mean it is not attached to RNA (which presumably binds the NiNTA column)? If my interpretation is correct, does that suggest that the majority of the bioluminescence signal in cells is not associated with RNA transcripts?

5) The authors answered my question in their rebuttal. But, as the authors note there are a number of issues related to kinetics of detection, delivery kinetics, etc that are not addressed in this experiment. Therefore, it seems the real value of Fig 6 is demonstrating that RNA lantern signal can be detected when implanted into a mouse (which does have value for the RNA field because I doubt the same could be said for the fluorescent reporters). But it is a stretch to say that lanterns can be used for RNA imaging in tissue since the authors simply implanted loaded cells into a mouse. I recommend editing the manuscript to tone down the claims from this experiment.

6) This comment has been addressed.

Finally, minor comments have been addressed.

Reviewer #3

(Remarks to the Author)

As far as I'm concerned, the authors have responded point by point to my various observations and I agree that the article should now be published.

Version 2:

Reviewer comments:

Reviewer #1

(Remarks to the Author)

The revised manuscript mostly addressed the reviewer's prior concerns though the data of RNA granules is not convincing. The manuscript will be acceptable in this journal.

Reviewer #2

(Remarks to the Author)

The authors have addressed my comments and concerns. I am supportive of publication.

Point-by-point response

We would like to thank the reviewers for their careful review of our work. The constructive comments and feedback helped us to improve the manuscript. Our point-by-point response is below.

Reviewer 1

This reviewer felt that the experiments are carefully designed, and in principle has many advantages over existing methods. However, all the characterization experiments were conducted in in vitro translation system (IVTT), and there were no information on the detection limit, dynamic range, or quantitiveness. Thus, while individual experiments and system optimization seem to be conducted appropriately, characterizations and applications were not sufficiently conducted to demonstrate the practical performance of the present system. In addition, several concerns listed below are raised that the authors need to be addressed.

We agree that the platform has many advantages over the current gold standards in the field. In addition to showcasing improved sensitivity with a much smaller epitope (Figure 4A-B in revised submission), we evaluated the tags and lanterns for imaging previously reported transcripts (Figures 5D, S10-11, S13 and associated movies). We also performed additional biochemical experiments to determine the overall detection limit and dynamic range (Figure 3). We further carried out titration and enrichment experiments in cells and IVTT mixtures to quantify RNA levels based on the photon outputs (Figure S9). To our knowledge, this is the first such measurement in the RNA field. Collectively, these data underscore the utility of the platform and its practical application to imaging RNA.

1. In RNA tracking, the bait would be used as a tag fused to the target RNA to detect and visualize the RNA. The authors should show more applicability to fuse the bait to different RNAs that are meaningful biologically to detect or visualize.

We have now used the RNA tag to image biologically relevant transcripts in cells, including *CDK6* and *β -actin* (Figures 5D, S11, S13, and associated movies). Both transcripts are known to localize to granules upon cellular stress. We also performed a dynamic imaging experiment with an inducible reporter system to visualize gene expression in real time (Figure S10). These examples confirm the utility of the RNA tags and lanterns for imaging relevant targets.

2. The title of this manuscript says “tracking,” but there were no experiments of tracking that required time-course analysis.

This is a good point, and as noted above, we have now provided direct examples of dynamic imaging in live cells (Figures 5D, S10-11, S13, and Supplementary Movies 1-6). We further measured the kinetics of lantern formation (Figure 3). Robust photon production was observed within minutes of RNA introduction, allowing RNA tracing with temporal resolution on par with the most optimal split fluorescent protein reporters (and at least an order of magnitude faster than conventional fluorescent proteins).

3. The performance of the RNA detection and visualization system can be expressed by such parameters as the detection range and limit of the target RNA concentration, which are calculated based on the concentration of the bait RNA and the lantern proteins. However, the characterization experiments were performed only in IVTT, and no RNA bait and lantern concentration estimations were conducted. Therefore, the performance of this system cannot be quantitatively evaluated.

In the original submission, we used IVTT analyses to estimate the limits of detection and verify bioluminescent complex formation in vitro (Figure 2 in the revised manuscript). We have now performed similar experiments in whole cells. As shown in Figure S9, we observed about one-tenth the concentration of mRNA in cell lysate compared to plasmid DNA introduced into the cells. The signal scaled approximately linearly between 10–300 ng DNA template. About 1% of the tagged RNAs were retrievable under these conditions, and titration of the target RNA in cell extracts implied that the protein components were expressed around nanomolar concentrations. We further measured the kinetics of lantern assembly in vitro (Figure 3). Luminescence was detected within 30 s upon RNA (nM) addition, with peak luminescence reached within 20 min. These data indicate the sensitivity of the platform and provide a benchmark for the RNA field. It should also be noted that if even more sensitive detection is required, longer acquisition times can be used.

4. All the bar graphs other than those in Fig. 4 are accompanied by error bars that depict the standard error of the mean, but the data points in each bar graph have large deviations. As the deviations in bioluminescence intensities would often be inevitable, the degree of the deviations should be evaluated by showing standard deviations to evaluate the quantitative capability of this method. Especially the data in Fig. 3G highly deviate, and it is important to show that there are significant differences to conclude that M-5-P had the strongest luminescence among the RNA baits.

We thank the reviewer for this comment. We updated the text to include standard deviation measurements (SDM) where appropriate. We also performed additional statistical analyses (via unpaired t-tests), and provided *p*-values for relevant comparisons.

5. Although noticed in the brackets, the expression of three types of SmBiT (high/99, low/114, and ultra/86) needs to be consistent between the main text and the legend of Fig. S1.

We have updated the caption for Figure S1 to be consistent with the naming scheme (SmBiT^{high}, SmBiT^{low}, and SmBiT^{ultra}) throughout the manuscript.

6. In Fig. S1B, many potential readers of this article may not be familiar with the word "Retic". The reviewer recommends revising it to reticulocyte, as written in the main text.

We have updated the plot to include "RRL" to match the abbreviation given in the caption.

7. The bars of SmBiT low and 3x-SmBiT low in Fig. S1B seem identical. It is, of course, OK if the values are accidentally quite similar, but the authors should check for the data mix-ups.

We thank the reviewer for alerting us to this plot. We double checked the raw data and confirmed that our original submission was accurate. We have since included the individual data points for each replicate, for improved clarity.

Reviewer 2

Here, the authors expand the utility of this system with two significant advances: 1) they fuse components of NanoBiT (smBiT and IgBiT) to MCP and PP7 and optimize the fusions, and 2) they combine the MS2 and PP7 stem loops into a single optimized RNA bait. The authors demonstrate that a single RNA bait gives rise to bioluminescence signal above background in vitro, in cells, and in vivo. This is an exciting development and an important addition to the RNA imaging toolbox

We thank the reviewer for this high praise!

But the field of RNA imaging is advancing quickly. Many flashy preliminary reports turn out to be poor tools (e.g. performance of peppers and mango in original reports versus in Bühler et al, Nature Chem Bio, 2023). Therefore, while the RNA lantern tool represents an exciting advance, there are some important benchmarking experiments that should be done to provide the field with a more comprehensive analysis of this exciting new tool. Upon completion of these experiments, I would be supportive of publication in Nature communications.

We agree with the need for careful benchmarking and analysis, and we have performed additional experiments to confirm the robustness of the platform. Specific details are provided below.

1) Characterization of RNA detection in cells is too preliminary and needs to be expanded. In Fig 4B, 4 cells are visible in the GFP channel, whereas only 2 cells are visible in the luminescence channel (the same is true in images in Fig S7); this seems problematic and suggests the correlation between RNA expression (as defined by GFP signal) and bioluminescence signal needs to be defined much more rigorously (The Pearson's coefficient is a population average and not terribly informative).

We apologize for the lack of clarity in the original submission. We did not intend to draw a quantitative comparison between the two reporters; there is no reason that such a correlation must exist. We merely used the fluorescence signal to gauge successful transfection and select cells for subsequent luminescence measurements. We would expect luminescence to only be observed in cells expressing GFP (based on the construct design), and this is clearly evident in both widefield (Figure S8) and magnified views of the cells (Figures 5 and S7). Several fluorescent cells are also devoid of luminescence, and this

is to be expected based on the hook effect and time-dependent nature of the signal. Additional correlations between bioluminescent and fluorescent signal production are also showcased in the examples of model imaging (Figures 5D, S11, S13, and associated movies).

It is not clear what each data point in 4C represents (1 cell? 1 field of view? The average of a field of view from one experiment?) but the standard for imaging studies is that measurements need to be made on tens to hundreds of individual cells. Given the apparent variability in 4B and S7, the authors should not only report the total bioluminescent signal in individual cells for cells expressing the bait versus those expressing a non-binding transfected RNA (like GFP). But they should also report what fraction of cells that show a GFP signal above background also show a bioluminescent signal (for both the bait-expressing and control cells).

We apologize for the confusion. The quantified data (now Figure 5C) derived from a bulk cell measurement using a luminometer. Flow cytometry was used to control for the expression level of each reporter. We have also added relevant information on the fraction of bioluminescent cells in the samples shown in Figures S7-S8.

2) The structured RNA bait is a clever idea and a valuable contribution to the field. However, for many users the goal of RNA imaging is to study RNA biology. It seems important to provide some basic measurements on whether the structured bait (and the binding of the lantern to the bait) alters RNA localization or stability.

We agree and have now used the RNA tag to image biologically relevant transcripts in cells, including CDK6 and β -actin (Figures 5D, S11, S13, and associated movies). Both transcripts are known to localize to granules upon cellular stress. Our imaging data corroborate these findings. There will, of course, be instances where the tag and bound complex remain too large (and thus perturb) the target transcript. Such cases will need to be determined empirically, though, like for all imaging reporters. Our platform has a significant advantage over existing methods, based on the much smaller size of the components. So, it is likely to enable applications where other tools have failed.

For example, it has been shown that the MS2 stem loop interaction with MCP stabilizes RNA transcripts and hence alters the half-life of RNA decay. This could be done by smFISH (for localization) or RT-qPCR for half-life. Similarly, it seems important that the authors characterize the relationship between RNA decay and BL signal decay. What happens when the tagged RNA decays? Does NanoBit remain reconstituted and competent to emit photons? As noted by the authors, there is intense interest in tracing the lifecycle of RNAs in mammalian cells and animals. This requires a tool that emits a signal when the RNA appears and stops emitting the signal when the RNA decays.

We agree that tracing the entire lifecycle of RNAs is an exciting pursuit. This work marks a significant advance in the initial phase, reporting on RNA generation. We are currently developing probes to specifically track RNA decay (where signal *increases* upon RNA degradation), but this work is beyond the scope of the manuscript. To address the immediate concern of whether bioluminescent signal tracks with the location and abundance of target transcripts, we analyzed transcript production in live cells (current Figure S10) and measured the kinetics of the complex assembly in vitro (Figure 3). The lantern signal in vitro was also depleted upon RNase treatment (data not shown).

Both in vitro and in live cells, we observe a decrease in signal on 30–60 min time scales, but at this point, we do not know whether this represents RNA decay, lantern turnover, or luciferase substrate depletion. As the reviewer noted, MS2 is known to stabilize RNAs; therefore, sensitive experiments measuring RNA decay will require a separate line of studies in cells. These points have also been added to the main text discussion.

3) What are the kinetics of reconstitution? The IVTT seems like it would be the perfect system to define this because you could look at the appearance of luminescence signal after production of the M-3-P RNA bait. It seems this would be important to define because some reconstitution systems can be slow (for example split-GFP 1-10/11 can take 30-60 min for fluorescence signal reconstitutions). This would be important for potential users to evaluate when after export to the cytoplasm and RNA of interest would be visible.

We thank the reviewer for this suggestion. We have now completed a time course experiment (shown in Figure S10) for RNA expression in live cells. We further measured the kinetics of lantern assembly in vitro (Figure 3). Signal was immediately detected using upon 1 nM RNA addition (by the first time point was collected at 30 s), with half-maximum luminescence observed in 4 min and full luminescence reached within 18–20 min.

4) *The data in 2C are very important/valuable to define the detection sensitivity. The fold change in BL signal over no RNA control is 20x, whereas in vitro the turn on is 300x. This is not surprising as most fluorogenic aptamers exhibit a diminution in signal-to-background in cells compared to in vitro. But it would be valuable for the authors to explicitly discuss the decrease and speculate on the reasons. Is this related to the expression level of the lantern in cells. Since each component of the lantern has an affinity tag, is it possible for the authors to quantify how much of the lantern is expressed in their stable cell line?*

We performed a comprehensive analysis to address this question. As shown in Figure S9, we observed about one-tenth the concentration of mRNA in cell lysate, compared with plasmid DNA introduced into cell culture media. The signal scaled approximately linearly across a range of DNA template concentrations. About 1% of the tagged RNAs were retrievable under these conditions.

One reason for the lower fold-change is the heterogeneous expression of the protein and RNA components in individual cells, likely producing ratios of RNA-to-protein at levels where hook-dependent effects are observed.

5) *For the in vivo proof of concept experiment: why were the cells incubated with luciferin prior to implantation? It seems unlikely that users interested in tracking RNA in vivo would follow this experimental paradigm. Why not implant the cells, then deliver luciferin to the mouse? The experimental design seems particularly contrived.*

In the initial application, we wanted to avoid substrate delivery kinetics as a variable in signal output. A similar procedure was used in a recent report evaluating a luminescent RNA reporter (Ref. #8). Our main objective was to showcase that RNA dependent signal could be observed through tissue, and this was clearly established. We are now performing more comprehensive in vivo studies with brain-localized reporters, and these results will be published in due course.

6) *For the BRET experiments in Fig 3G: Isn't the fluorescent output what matters here with respect to signal turn on (reconstitution of NaoBiT leads to energy transfer to the fluorescent protein)? What is the yellow signal and red signal above the M-3-Pmut or no RNA control?*

These data were collected without filters (now Figure 4G), and the signal can be attributed to substrate autooxidation and/or LgBiT itself, which can minimally turn over the substrate. Both of these scenarios result in low levels of background photon production.

The reviewer also requested some minor edits:

1. *Fig 3B – Here the M-3-P only shows ~80x BL signal over no RNA control whereas in 1E it shows 300x? What is the reason for the variability?*

We attribute much of the variance to the composition of the lysate samples, which vary by lot in terms of ion concentrations and other factors known to impact RNA structures and luciferase assembly.

2. *The data in Fig 2C are valuable for defining detection sensitivity. It would be valuable for potential users to put these RNA concentrations in context. What is the typical expression of a low, medium, highly expressed RNA?*

We have since performed additional experiments to examine the limit of sensitivity in cellular settings. These data are shown in Figure S9. Given sufficient imaging time, it is possible to see single enzymes. For dynamic imaging studies (where seconds-to-minutes acquisition is required for temporal resolution), though, we can observe 1 nM complexes in vitro in 30 s. We anticipate similar photon outputs would be observed in cell culture with the same number of complexes. RNA expression levels in vivo are highly variable, ranging from pM (1 molecule per small—10 μm —

eukaryotic cell) to μM levels. The data in Figure S9 show that DNA transfection yields almost 10 nM mRNA in HEK293 cells.

3. Page 7, please describe what you mean by “hook effect”

This term is commonly used in one-step binding and detection assays (e.g., ELISAs) when large amounts of the target result in artificially low or decreased signal. In our case, large amounts of RNA result in luciferase fragments bound to separate transcripts, resulting in diminished photon production. Both fragments must be bound to the *same* target for signal to be produced.

Reviewer 3

While another group has recently explored the use of split luciferase to visualise targeted RNA, but with poor performance, J. Prescher and his co-authors have developed an alternative system with much better performance. The design is ingenious, albeit intricate. It offers the potential to explore a broad spectrum of transcripts, providing a promising route for enhancing our understanding of RNA biology in vivo. While the manuscript is well-written and the authors' conclusions are generally supported by the data, some major and minor points need addressing before the paper is suitable for publication in Nature Communications. We thank the reviewer for the high praise and recognition of the potential utility. We have addressed the specific concerns below.

Major points

The literature shows a growing interest in the challenging visualization of transcripts in their real environment. It is difficult to assess the performance of this new system without a suitable benchmark. It would be useful to compare this new system with the unsplit NanoBiT system. This control is missing.

While we understand the concern, the proposed experiment is confounding, as the unbound luciferase is always “on”, regardless of whether the RNA tag is present or not. We have performed benchmarking experiments, though, to address the reviewer’s concern. These include pull down assays to directly confirm target engagement and reporter assembly (Figures 2 and S9), “add back” experiments to gauge the number of unbound smBiT and LgBiT fragments in a given assay (Figure S1), and kinetics assays (Figure 3).

For real-time detection, it is important to determine the signal intensity as a function of time in order to know the time window for use. Time-dependent bioluminescence intensity experiments are missing. What is the duration of the signal in relation to unsplit NanoBiT?

This is a good point, and we have now directly measured signal turn-on in the presence of target RNAs. As shown in Figure 2, bioluminescence is observed within minutes. This is in contrast to many fluorescent protein-based methods that can take hours, due to extra time needed for chromophore maturation. The direct experimental comparison to unsplit NanoBiT is ambiguous in our case, as signal from the enzyme is observed whether the RNA tag is present or not. However, the light output from related split NanoLuc applications has been found to compare favorably to NanoLuc itself (e.g. within an order of magnitude intensity-wise, and with similar duration, *ACS Chem. Biol.* **2016**, *11*, 400).

The notion of small size is relative. The TAG used is 69 nucleotides long, i.e. approximately 23 kDa, which could pose problems when introduced in small non-coding RNAs. The term small should be deleted from the abstract. What impact does the tag have on transcript expression and functionality?

The reviewer is correct in noting that “small size” is only a relative term. Like every tag and chemical probe, the impact on target expression and functionality will be context dependent. We are showcasing model systems in this manuscript, but the tag is significantly smaller than gold standards in the field (which often use 24 copies of the MS2 and PP7 aptamers and linkers between them) and thus an important advance. Recognizing that this platform will not be suitable for all RNAs, though, we are working on even smaller variants. We updated the abstract to be more precise regarding the tag size parameter.

Minor points

The bibliography should be updated to include the review by Palmer et al. published in Cell Chemical Biology on September 17, 2020.

We agree, and have since updated the references to include this excellent review article.

*Page 7: “were observed with the additional G4S units compared to the original lanterns both in vitro and”
This sentence is incomplete*

We apologize for the typo. The sentence has since been updated.

Page 16, last line: The results of at least 3 experiments are needed to calculate a standard deviation for (D).

We thank the reviewer for catching this error. We have since clarified the number of replicates for each experiment and provided standard deviations where possible and/or appropriate. In the few cases where only two replicates were performed, we provide either standard errors of the mean or an average value.

The way in which the signal intensity is calculated is not specified in the experimental section. For example, it is not clear what bandwidth (filter), acquisition time or equipment is used. These details are missing from the experimental section.

We apologize for the lack of clarity regarding the imaging parameters. We have since updated the methods section and reporting summary to include more details.

The structure of luciferins and their maximum emission should be given in SI.

Furimazine (Fz) was used for all but one study in this manuscript. For clarity, we have added the structure to Figure 1 and noted its emission wavelength in the caption. D-Luciferin was used for a single experiment (Figure 4, C-D). The emission wavelength for this substrate was also added to the caption.

Point-by-point response

We thank the reviewers again for their careful review of our work. Our point-by-point response is below.

Reviewer 1

The authors have revised the manuscript sufficiently for the most part, adding experimental data in response to the reviewer's comment. However, for some additional data it is unclear which part of the data to focus on to understand the authors' argument. Improvements are needed in the presentation of the data. We appreciate this comment and have revised the manuscript for improved clarity.

1. *In response to comment 1, the authors performed experiments to monitor beta-actin mRNA and CDK6, which were concentrated in the RNA granules when the cell was stressed, as mentioned in the manuscript. However, it is not clear from the microscopic images shown in Figure 5D whether the RNA granule marker (GFP-G3BP1) and luminescence (RNA) are merged well even in the enlarged images 60 min after stress application. The authors should compare the localisation of the RNA granule marker and the luminescence signals before and after stress addition on multiple cells (it is natural that some cells do not form stress granules; overexpression the granule marker may make it difficult to visualize precisely the RNA granules) and then perform the quantitative analysis of the colocalisation such as Pearson's correlation coefficient. In addition, it may be reader-friendly if the authors add some arrowheads to the images to indicate the colocalised areas.*

We thank the reviewer for raising these concerns. Assessing co-localization via the merger of fluorescent and bioluminescent images is challenging, as it is not possible to rapidly switch between the modalities and acquire sufficient signal for an overlay. Long acquisitions are required for bioluminescence, complicating the assignment of subcellular features (and, in this case, target transcripts). For this reason, we are hesitant to provide traditional stats on signal co-localization for subcellular features. Co-localization measures on the whole cell level (as in Figure 5B) are more feasible. Indeed, it is rare to see mergers of fluorescence and bioluminescence images in the published literature. (Note recent works from leading laboratories: *PNAS*. **2015**, 112, 4352, *Sci. Rep.* **2018**, 8, 8984, *ACS Sensors*. **2019**, 4, 7, *Sensors. Basal*. **2019**, 16, 3502, *Nat. Comm.* **2022**, 13, 3967, *Nat. Methods*. **2023**, 20, 1563, *ACS Sensors*. **2023**, 8, 11 where only correlations at the whole-cell level are made, if any).

To avoid any confusion and over-interpretation of our data, we modified the text to discuss the images in Figure 5D in qualitative terms only. We also added more discussion on the current limitations of bioluminescence microscopy, including the long acquisition times which hinder conventional overlays and other analyses.

Reviewer 2

This resubmission is an improvement. However there are a number of previous concerns which were not adequately addressed (see below). I remain supportive of publication of this work in Nature communications if the comments outlined below can be addressed. This largely involves editing the manuscript to more clearly articulate the strengths and limitations of the tool and acknowledge important experimental factors that have not yet been defined because they are beyond the scope of the manuscript. We thank the reviewer for careful analysis of the revised manuscript. We have addressed the additional concerns, and our comments are below.

1) *I think the concern articulated in my first comment was misinterpreted. I understand that GFP signal was used to identify transfected cells; my concern is that it seems only a subset of the transfected cells show luminescence signal suggesting poor detection of RNA in cells. The manuscript text is misleading on this point and implies that transfected cells show luminescence signal. However, Figure S7 now presents a quantification of the correlation and it is only 62%. I think this needs to be flushed out and addressed directly in the manuscript. It seems important for potential users of this technology to know how well the tool performs at the single cell level. The authors should quantify the correlation (what % of GFP+ cells also show luminescence signal), report this in the main manuscript, and discuss the limitations of the technology that influence this value. For example, in the rebuttal the authors attribute this performance to*

the “hook effect” and “time-dependent nature of the signal”. The hook effect is an important consideration as the performance of the tool (i.e. its ability to detect RNA) diminishes at high and low RNA concentrations. This will directly impact the type of RNA transfect that the tool can effectively detect and the authors should discuss what types of transcripts are likely to be able to be detected by RNA lanterns.

We apologize for misinterpreting the previous critique. As noted in response to Reviewer 1, the microscopy data are difficult to generate easily, as the instrumentation does not support seamless switching between fluorescence and bioluminescence readouts. We have updated the text to draw attention to the concern.

The value reported in Figure S7 (62%) was the percentage of GFP+ cells that show luminescence signal. We have now included this result in the main text, to draw attention to the variable. It should be noted, though, that the value is highly context dependent and not prognostic of how well the probe set would perform in any given case (i.e., not all single cell experiments will have similar target levels or constraints.)

I am not sure what the authors mean by “time-dependent nature of the signal” (the fact that the luminescence decays or that detection sensitivity depends on integration time). Either way, if the time-dependent nature of the signal (which is not a feature of fluorescent reporters) represents a limitation of the tool, again this should be discussed in a way that helps potential users identify appropriate experimental “fits” for this tool. Also, although the authors state they quantify the fraction of bioluminescent cells in Fig S7 and S8, they only present a quantification in S7.

We apologize for the lack of clarity. In the earlier submission, “time-dependent nature of the signal” was meant to indicate that luminescence decays over time, in the absence of additional substrate. We have since modified the text to highlight this limitation and other parameters impacting the overall signal. We also note that use of slow-release luciferins can circumvent some of these limitations, and will be the subject of ongoing work.

2) Unfortunately, the new data (Figure 5, Supplementary Figure 11 - 13, videos) presented on localization of transcripts to stress granules are not convincing. Either due to low magnification, poor signal to noise, or low resolution, stress granules are not evident in images or movies. The only image that shows something that *could* be construed as stress granules was cell #3 in Figure S13, but unfortunately the strangely punctate signal was present before addition of arsenite, so can't be attributed to stress granules. None of the new data show colocalization of RNA lanterns with G3BP1 in stress granules. Therefore, I disagree with the authors' conclusion that “transcript localization to stress granules was observed”. Further, there is no quantification of recruitment/localization (what fraction of stress granules show transcript localization) and there is no negative control where (for example) cells transfected with the mutant M-X-P show no colocalization with stress granules). I understand the challenge in showing stress granules using bioluminescence. In my original comment, I was not suggesting the authors try to track transcript movement at the subcellular level. This is not an inherent strength of bioluminescence. In fact, it is a weakness compared to fluorescence. Rather, bioluminescence is particularly valuable for animal imaging. In my original comment, I was trying to express concern that the structured aptamer could alter the properties of the RNA transcript to which it is attached, in particular the half-life of the RNA (because the MS2 system has previously been shown to alter mRNA half-life). This could have been addressed by comparing the expression level of XFP mRNA with and without the M-3-P tag. In the revised manuscript, the authors have not addressed whether the M-3-P tag perturbs the transcript to which it is bound. In that case, I would recommend the authors add to the discussion that future experiments should be conducted to determine whether the structured tag perturbs RNA lifetime. As for the stress granule experiments, I don't think the authors' claims or conclusions are supported by the data presented. That said, I don't think they are important for the manuscript (users would be unlikely to use RNA lanterns for subcellular RNA detection) so I would recommend removing them.

We appreciate the feedback here, and the need for careful validation with all imaging tags. It is definitely possible for the structured RNA to perturb localization and alter (likely increase) the half-life of the mRNA, and this would need to be checked on a case-by-case basis. We have now added a precautionary note along these lines in the main text. As noted above (and in response to Reviewer 1), we have also toned down claims about exact co-localization, as only general trends can be gleaned from our approach due to

current limitations in bioluminescence microscopy. The videos are still included in this submission, as they provide the end user with a visual representation of the imaging output.

In my original question, I asked whether the bioluminescence signal correlated with the presence of RNA in cells (i.e. if the RNA decays does the bioluminescence signal disappear). I understand the authors rebuttal that the kinetics of the signal is complicated as it depends on RNA lifetime, lantern lifetime, and bioluminescence decay. It would be good for the authors to systematically and rigorously characterize this in future publications. But I would appreciate it if the authors could provide an explanation for the observation in Figure S9B that the majority of the bioluminescence signal doesn't "pull down" with the RNA (i.e. flows through and doesn't bind to the NiNTA column suggesting it isn't associated with the NiNTA aptamer in the tag).

We apologize for not fully addressing the concern in the prior round of review. We have since added more discussion on the need to analyze how well the signal tracks with RNA decay in future studies. Regarding the data in Figure S9, we hypothesized that the large amount of uncaptured complex (represented by signal in the flow-through) was due to bead saturation or otherwise poor capture. The Ni-NTA aptamer is small and only a single copy is present per target transcript, which may occlude it from easily binding the beads. Smaller constructs bind the Ni-NTA beads efficiently in vitro (note Figure 2), suggesting that the affinity and off-rate are not limiting for the pull-down, rather, the much larger mRNA construct pulled-down from cell extracts appears to bind less efficiently. As expected, when additional beads were added to the retrieval assays, more target transcript was bound (Figure R1, below), suggesting that the Ni-NTA aptamer is active and associated with the bioluminescent complex.

Figure R1. Improved capture with increasing amounts of target and beads. (A) Cells expressing RNA lanterns were transfected with DNA encoding *GFP-M-3-P* (or no DNA). Lysates were incubated with either HisPur™ Ni-NTA magnetic beads (1 or 10 μL) and analyzed as in Figure S9. (A) Bioluminescence images of complexes captured from homogenized cells. (B) Quantification of photon outputs from the samples in (A).

3) The authors have addressed my question with very nice new data presented in Figure 3. Thank you for the kind feedback.

4) The authors addressed my question in the rebuttal but did not sufficiently address it in the manuscript. This could be remedied by straightforward editing of the manuscript. The upshot seems to be that detection depends on RNA concentration, concentration of the protein components, and also probably some

complicated kinetics with respect to RNA lifetime and bioluminescence lifetime. None of these caveats are explicitly mentioned in the text. When the authors discuss performance in cells, they should directly address these factors. One thing I don't fully understand about the Supplementary 9 data: in 9B the majority of the luminescence signal comes through in the flow through. Does that mean it is not attached to RNA (which presumably binds the Ni-NTA column)? If my interpretation is correct, does that suggest that the majority of the bioluminescence signal in cells is not associated with RNA transcripts?

We apologize for not making the points clear in the prior submission. We have since edited the manuscript to draw attention to variables impacting light output. The questions raised concerning Figure S9 were addressed in the response to point 2 above.

5) The authors answered my question in their rebuttal. But, as the authors note there are a number of issues related to kinetics of detection, delivery kinetics, etc that are not addressed in this experiment. Therefore, it seems the real value of Fig 6 is demonstrating that RNA lantern signal can be detected when implanted into a mouse (which does have value for the RNA field because I doubt the same could be said for the fluorescent reporters). But it is stretch to say that lanterns can be used for RNA imaging in tissue since the authors simply implanted loaded cells into a mouse. I recommend editing the manuscript to tone down the claims from this experiment.

We appreciate this point, and have updated the text accordingly.

6) This comment has been addressed.

We thank the reviewer again for carefully reviewing our revised manuscript.

Reviewer 3

As far as I'm concerned, the authors have responded point by point to my various observations and I agree that the article should now be published.

We thank the reviewer for the feedback and recommendation to publish.

Point-by-point response

We thank the reviewers again for their careful review of our work. Our point-by-point response is below.

Reviewer 1

The revised manuscript mostly addressed the reviewer's prior concerns though the data of RNA granules is not convincing. The manuscript will be acceptable in this journal.

We appreciate this comment and have modified the text to avoid any over-interpretation of the data. As noted in the last round of review, definitive co-localization (“ground truth”) studies are challenging owing to the long acquisition times required.

The text has been updated to reflect this point:

Punctate-like structures were observed in some cells (**Fig. 5D, Supplementary Fig. 11B, Supplementary Fig. 12**)³⁵⁻³⁸. **However, definitive granule localization could not be confirmed in any case.** This is perhaps due to the longer acquisition times required for bioluminescence, complicating the assignment of subcellular features in relation to fluorescent markers (G3BP1, in this case).

Reviewer 2

The authors have addressed my comments and concerns. I am supportive of publication.

We thank the reviewer for the feedback and recommendation to publish.